# Selective interactions at pre-replication complexes categorize baseline and dormant origins

Bhushan L. Thakur [1], Christophe E. Redon[1], Haiqing Fu [1], Robin Sebastian [1], Nana A. Kusi[1], Sophie Z. Zhuang[1], Lorinc S. Pongor [1,3], Vilhelm A. Bohr [2] & Mirit I. Aladjem [1]

DNA synthesis in metazoans initiates within a select group of replication origins (baseline origins), whereas other (dormant) origins do not initiate replication despite recruiting apparently indistinguishable pre-replication complexes. Dormant origins are activated as backups when DNA synthesis stalls, allowing for complete genome duplication, yet it is unclear how cells selectively differentiate between baseline and dormant origins. We report here that during unperturbed cell proliferation, dormant origins selectively bind phosphorylated RecQL4 (pRecQL4), a member of the RecQ helicase family mutated in Rothmund-Thomson, RAPADILINO and Baller-Gerold syndromes. Origin-bound pRecQL4 prevents the binding of an essential replication initiation complex, MTBP-TICRR/TRESLIN, to dormant origins, thus restricting replication initiation to baseline origins. When cells encounter replication stress, pRecQL4 is required for the dissociation of the MTBP-TICRR/TRESLIN complex from chromatin, which, in turn, facilitates the subsequent redistribution of MTBP-TICRR/TRESLIN to both baseline and dormant origins and allows recovery from replication inhibition. Thus, the interactions between the MTBP-TICRR/TRESLIN complex and pRecQL4 at replication origins are critical for replication origin choice and facilitate recovery from replication stress.

In eukaryotic cells, chromosome duplication requires the timely recruitment of pre-replication complexes (pre-RCs) to chromatin during the G1 phase of the cell cycle. Mammalian pre-RCs assemble in excess at thousands of chromosomal sites, known as replication origins, and a select fraction of those origins (termed baseline origins) initiate replication[1–3]. Another group of origins recruits pre-RCs, but do not initiate replication during unperturbed cell proliferation (dormant origins). Pre-RCs bound to dormant origins can potentially be activated to complete genome duplication when replication from baseline origins halts or slows. However, in normally growing cells, the initiation of DNA replication from dormant origins is constrained to prevent transcription–replication collisions, gene amplification and DNA

breaks[4–8]. The selective, sequential, and often tissue-specific activation of baseline replication origins facilitates the transfer of accurate genetic and epigenetic information to daughter cells. Conjointly, the limited, replication stress-induced activation of dormant origins is a critical compensatory mechanism that serves to complete the duplication of the entire genome under potentially genotoxic conditions, limiting mutations, translocations and other consequences of genotoxic lesions[5–7].

To assemble the pre-RCs, an origin recognition complex (ORC), along with the adaptor proteins CDC6 and CDT1, recruits an inactive form of the replicative helicase, minichromosome maintenance 2–7 (MCM) complex. At baseline origins, pre-RCs are converted at the G1-S

[1]Developmental Therapeutics Branch, Center for Cancer Research, NCI, NIH, Maryland, MD, USA. [2]Department of ICMM, University of Copenhagen, Copenhagen, Denmark. [3]Present address: Cancer Genomics and Epigenetics Research Group, HCEMM, Szeged, Hungary. ✉e-mail: aladjemm@mail.nih.gov

transition to pre-initiation complexes (pre-ICs)[2,9,10] that subsequently form active helicase complexes. The transition from pre-RC to pre-IC requires the binding of additional components, including DNA topoisomerase II binding protein 1 (TOPBP1), TOPBP1 interacting checkpoint and replication regulator (TICRR/Treslin), Mdm2-binding protein (MTBP), Sld5, Psf1, Psf2, and Psf3 (GINS 1–4 complex) and CDC45, along with the phosphorylation activity of two kinases: DBF4-dependent kinase (DDK) and cyclin-dependent kinase 1 or 2 (CDK1/2). DDK phosphorylates the MCM helicase at several residues[11–14], including an essential phosphorylation on serine 139 of the helicase subunit MCM2 (pMCM2-S139). Our previous studies have shown that DDK-phosphorylated MCM2 on serine S139 (pS139-MCM2) preferentially associates with baseline origins, and is essential for the initiation of DNA replication[15], whereas other residues did not exhibit preferential phosphorylation at baseline origins. CDK1/2 phosphorylates TICRR/Treslin[16,17] and its partner MTBP[18], subsequently facilitating the recruitment of the MTBP-TICRR/TRESLIN complex to chromatin[17,19–22]. The recruitment of the MTBP-TICRR/TRESLIN complex, in turn, facilitates the integration of additional helicase components, CDC45 and GINS, in the MCM complex to form an active helicase that initiates replication[19,20,23,24].

RecQL4, an ortholog of the yeast pre-RC component SLD2, is a member of the RecQ helicase family, a group of proteins that are crucial for maintaining genomic stability. Metazoan RecQL4, which contains an SLD2 domain, can colocalize with replication origins and interact with pre-RC components, including phosphorylated MCM2[16,25–29]. Mutations in the RecQL4 gene are linked to several rare genetic disorders, including Rothmund-Thomson syndrome, RAPADILINO syndrome, and Baller-Gerold syndrome. These illnesses are characterized by defects in skeletal development, premature aging, and predisposition to cancer[30]. RecQL4 is a target of both CDK1/2 and DNAPK, including a CDK1/2-mediated phosphorylation at serines 89 and 251[31,32]. RecQL4 binds replisome components[29,33] and interacts with G-quadruplex sequences[34,35], which are implicated in origin activity and telomere maintenance. Tethering RecQL4 to pre-replication complexes enhances replication initiation at some origins[28], and in single molecule studies, RecQL4 acts downstream of one of the SLD2 mammalian orthologues, DONSON, to convert the two MCM complexes into separate replication forks[36]. Although RecQL4 is essential for replication origin activation in Xenopus egg extracts and chicken DT40 cells, in other cases it was reported to play a negative role in the initiation of DNA replication[26,28,36–38].

SIRT1 (the mammalian ortholog 1 of yeast Silent Information Regulator 2, a protein deacylase) and ATR (Ataxia-Telangiectasia and Rad3-related protein, a serine/threonine kinase) are central to the selective switch that suppresses replication initiation at dormant origins during unperturbed growth while activating dormant origins during replication stress. During unperturbed cell proliferation, SIRT1 prevents the activation of dormant origins by deacetylating TOPBP1, a pre-IC component and an ATR activator, thus preventing ATR recruitment to chromatin[15,39–41]. Conversely, when cells encounter replication stress, ATR activity increases, and it binds chromatin[42] to initiate a signaling cascade that phosphorylates serine 108 on MCM2 (pMCM2-S108) at dormant origins[15,43]. MCM2-S108 phosphorylation, in turn, permits DDK-mediated phosphorylation at MCM2-S139 at dormant origins, thus promoting replication initiation and allowing the completion of DNA synthesis adjacent to stalled replication forks[15]. By preventing ATR-mediated phosphorylation on MCM2-S108 and the subsequent initiation of DNA replication at dormant origins, SIRT1 keeps dormant origins inactive, thereby lowering the prevalence of deleterious lesions stemming from over-replication, including R-loops that reflect replication-transcription collisions, extrachromosomal elements and DNA breaks[15,44].

Here, we investigated the mechanistic basis of dormant origin dynamics using cells that harbor either intact or mutant SIRT1, which suppress or activate dormant origins. We first tested the interactions of MTBP, TICRR/Treslin and RecQL4 at replication origins during unperturbed growth and in response to replication perturbations. We found that during unperturbed growth, cells with intact SIRT1 exhibited a selective association of MTBP and TICRR/Treslin with baseline origins. Conversely, the CDK1/2-phosphorylated form of RecQL4 (phosphorylated on Serine 89; pRecQL4-S89) selectively associated with dormant origins. In cells with inactive SIRT1, in which origin dormancy was not evident, MTBP and TICRR/Treslin were associated with all origins. Next, we tested the effects of replication stress on these interactions. Upon perturbation of DNA replication in cells with active SIRT1, MTBP dissociated from baseline origins, and subsequently re-associated with both baseline and dormant origins to complete genome duplication. The re-distribution of MTBP, and the complete recovery from replication stress, required the phosphorylation of RecQL4-S89. These findings suggest that the selective associations of RecQL4 and MTBP with dormant and baseline origins underlie the orderly initiation of DNA replication during normal proliferation and facilitate the cellular response to replication stress.

## Results

### MTBP-TICRR/TRESLIN recruitment at active replication origins

To investigate the determinants of replication origin selection, we utilized stable HCT116 cells harboring SIRT1 variants that activate or suppress replication initiation at distinct subsets of replication origins[15,41]. These cells were all derived from a common parental cell line with a Null[SIRT1] background and expressed either active SIRT1 (WT[SIRT1]) or an enzymatically inactive H363Y-SIRT1 mutant (Mut[SIRT1]) (Fig. 1A, top panel)[15]. Both WT[SIRT1] and Mut[SIRT1] cells exhibited similar overall cell cycle phase distributions; however, a subset of Mut[SIRT1] cells displayed S-phase DNA content with reduced EdU incorporation (Supplementary Fig. 1A). We measured replication initiation activity in these cells by isolating short, RNA primed DNA strands (Nascent Strands) followed by sequencing (NS-seq)[15]. As shown in Fig. 1A, B, and consistent with our previous findings[15], almost all (92%) of the origins that initiated replication in the WT[SIRT1] cells also initiated replication in Mut[SIRT1] cells. In addition to the origins that initiated in both WT[SIRT1] and Mut[SIRT1] cells, cells harboring Mut[SIRT1] also initiated replication within an additional set of origins, which were inactive in WT[SIRT1] cells (Fig. 1B and Supplementary Fig. 1B–D). We classified the common replication initiation sites, which initiated replication in both WT[SIRT1] and Mut[SIRT1] cells, as baseline origins, whereas the additional replication initiation sites, which were only active in Mut[SIRT1] cells, were classified as dormant origins. As a control, we confirmed that dormant origins, inactive in WT[SIRT1] cells, were also activated under replication stress induced by aphidicolin (APH); APH-induced origins, activated in WT cells, exhibited 97.5% concordance with origins activated in Mut[SIRT1] cells, (Supplementary Fig. 1B–D).

We then determined if the pre-IC components MTBP, TICRR/Treslin, and RecQL4, which were reported to associate with pre-RCs prior to the initiation of DNA replication, selectively interacted with dormant and baseline origins. To assess if the binding of those proteins to chromatin required pre-RC assembly, we concomitantly measured chromatin binding in cells harboring an auxin (IAA)-inducible MCM2 degradation system (MCM2-mAID; Fig. 1C, D) with and without exposure to auxin. To assess if chromatin binding required DDK-mediated activation of pre-RC, we utilized a phospho-deficient mutant of MCM2-S139 (MCM2[S139A]) expressed in cells harboring the IAA-inducible MCM2 degradation system (MCM2-mAID; Fig. 1, C, D). We chose the phosphorylation of MCM2-S139 as a marker because, unlike the phosphorylation of other residues on MCM2 or other subunits of the MCM complex, DDK-mediated MCM2-S139 phosphorylation occurs preferentially at baseline origins and is essential for the initiation of DNA replication[15]. As shown previously, the extent of MCM2 phosphorylation at S139 can be accurately and reproducibly measured by

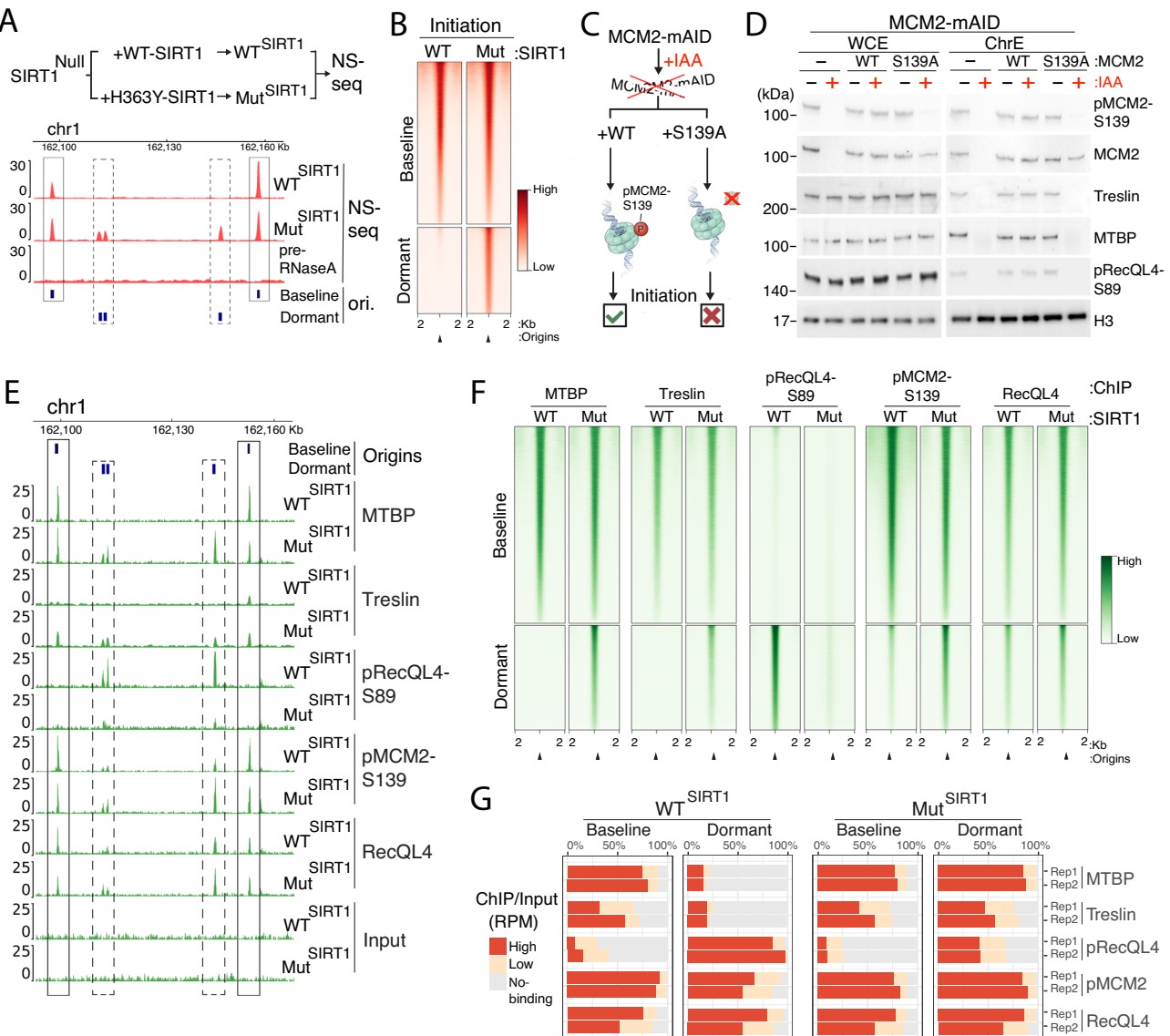

**Fig. 1 | Association of MTBP and phospho-RecQL4 with pre-RCs at baseline and dormant replication origins. A** Baseline and dormant origins. Top, experimental procedure. SIRT1 deficient (SIRT1[Null]) HCT116 cells were complemented with either WT-*SIRT1* (WT[SIRT1]) or *H363Y-SIRT1* (an inactive mutant; Mut[SIRT1]) cDNA[15]. Replication origins were mapped using nascent strand sequencing (NS-seq)[15]. Bottom, a genome browser view of NS-seq coverage across a representative genomic region on chromosome 1. Baseline origins (black rectangles) are defined as active in both WT[SIRT1] and Mut[SIRT1] cells, whereas dormant origins (dashed black rectangles) are active only in Mut[SIRT1] cells. Pre-RNase treatment serves as a control, as RNA primer removal allows lambda exonuclease to digest nascent DNA, eliminating origin peaks. **B** Nascent strand abundance (initiation) at baseline and dormant origins in cells harboring WT[SIRT1] (WT) or Mut[SIRT1] (Mut). **C** HCT116 cells containing auxin–inducible degron MCM2 (MCM2-mAID) were complemented with either intact (WT) or S139A substituted MCM2[15]. IAA (auxin, 500 μM for 16 h) depleted the endogenous MCM2-mAID. Complementation with MCM2-S139A, but not WT MCM2, inhibited replication initiation[15] (Supplementary Fig. 1E). **D** The abundance of pMCM2-S139, MCM2, Treslin, MTBP and pRecQL4-S89 in whole cell (WCE) and

chromatin extracts (ChrE) from HCT116 cells harboring MCM2-mAID complemented with MCM2-WT and MCM2-S139A with and without IAA. Representative of three independent replicates. The expression of MCM2-WT and MCM2-S139A was similar and comparable to the endogenous protein. **E, F** Binding sites of pMCM2-S139, MTBP, Treslin, RecQL4, and pRecQL4-S89 mapped by chromatin immunoprecipitation followed by sequencing (ChIP-Seq) in WT[SIRT1] and Mut[SIRT1] cells. pMCM2-S139 binding was measured in G1-synchronized cells, whereas binding of MTBP, Treslin, RecQL4, and pRecQL4-S89 was measured in asynchronous cells. **E** ChIP-seq coverage at the genomic region depicted in panel (**A**). From top, baseline and dormant origins; averaged ChIP-seq coverage (from two biological replicates) for MTBP, Treslin, pRecQL4-S89, pMCM2-S139, pMCM2-S139, RecQL4 and input controls from WT[SIRT1] and Mut[SIRT1]. **F** Heatmaps depicting chromatin binding at baseline and dormant replication origins in HCT116 cells harboring WT[SIRT1] or Mut[SIRT1] as measured as in panel (**E**). Baseline and dormant origins were stratified based on replication initiation activity as depicted in panels (**A**) and (**B**). **G** Quantification of chromatin binding patterns shown in panel (**F**) (above). Source Data are provided as a Source Data file.

immunoprecipitation, and chromatin binding sites of pMCM2-S139 can efficiently and reproducibly be mapped by chromatin immunoprecipitation followed by sequencing (ChIP-seq)[4,15].

As shown in Fig. 1D, addition of IAA resulted in MCM2 degradation in cells harboring the MCM2-mAID construct. Expression of MCM2 was evident in cells harboring MCM-mAID complemented with *WT-MCM2*,

and in cells harboring MCM-mAID with MCM2[S139A]. MCM2-mAID degradation led to the selective expression of the phospho-deficient form, MCM2[S139A] (which does not interact with the antibody that specifically recognizes the phosphorylated form, as confirmed previously[15,45]). Since MCM2-S139 phosphorylation is essential for the activation of replication origins[15], MCM2-mAID degradation followed

by the selective expression of the phospho-deficient form MCM2$^{S139A}$ prevented EdU incorporation (Supplementary Fig. 1E). As shown in Fig. 1D, addition of IAA in cells harboring MCM-mAID prevented the binding of MTBP and Treslin to chromatin, and binding was restored when IAA-mediated MCM2 depletion was complemented by WT-MCM2. In cells that harbored MCM2-mAID along with MCM2$^{S139A}$, chromatin binding of MTBP and Treslin was not restored after IAA-mediated MCM2 depletion. These observations suggested that the association of the MTBP-TICRR/TRESLIN complex with preRCs required chromatin harboring phosphorylated MCM2-S139.

We next asked if RecQL4, which was also reported to associate with pre-RCs, associated with chromatin in a pre-RC dependent manner. Because RecQL4 is phosphorylated by CDK1/2 at serine 89[31], we also asked if phosphorylated RecQL4 on serine 89 (pRecQL4-S89) would exhibit differential binding at replication origins. To evaluate the extent of pRecQL4-S89 chromatin binding, we developed a custom-made pRecQL4-S89 specific antibody. The antibody specifically recognized the phosphorylated form of RecQL4, and it did not interact with phosphatase-treated RecQL4 or with protein extracts from cells containing an engineered phosphorylation-deficient mutant RecQL4-S89A (Supplementary Fig. 1F). Although RecQL4 is a SIRT1 substrate[46], pRecQL4 levels were similar in cells harboring WT$^{SIRT1}$ and Mut$^{SIRT1}$ (Supplementary Fig. 1G).

Utilizing ChIP-Seq, we next mapped the chromatin binding sites of pMCM2-139, MTBP, TICRR/Treslin and pRecQL4-S89 (Fig. 1E, F). We then asked if pMCM2-139, MTBP, TICRR/Treslin and pRecQL4-S89 exhibited preferential binding to baseline origins (classified based on their activity in cells harboring WT$^{SIRT1}$ or Mut$^{SIRT1}$; Fig. 1A, B). We observed that MTBP, TICRR/Treslin and pMCM2-S139 predominantly colocalized with baseline origins in WT$^{SIRT1}$ cells and were associated with both baseline and dormant origins in cells harboring Mut$^{SIRT1}$ (Fig. 1E–G). RecQL4 was associated with all origins, including dormant origins, in both WT$^{SIRT1}$ and Mut$^{SIRT1}$ cells. pRecQL4-S89 preferentially colocalized with dormant origins in WT$^{SIRT1}$ cells (Fig. 1F, G). However, its association with dormant origins decreased in cells harboring Mut$^{SIRT1}$, in which dormant replication origins were capable of initiating DNA replication. The association of MTBP-TICRR/TRESLIN with active baseline origins, and the association of pRecQL4-S89 with inactive dormant origins, were not observed upon degradation of MCM2 in cells harboring MCM2-mAID alongside with the mutated MCM2-S139A (Supplementary Fig. 1H).

We next asked if the association of the MTBP-TICRR/TRESLIN complex with active origins was also detected in non-transformed cells. We have previously shown that pharmacological inhibition of SIRT1 activated dormant origins in primary and immortalized human fibroblasts[44]. We therefore mapped replication initiation sites as well as the binding sites for pMCM2-S139, MTBP and pRecQL4-S89 in immortalized fibroblasts from a healthy individual with and without exposure to Ex527, a SIRT1 inhibitor (Supplementary Fig. 1I). Due to limited sample quantities, we used antibodies directed against MTBP, as the MTBP antibody was more efficient than the TICRR/Treslin antibody. Consistent with previous observations[15,44], exposure to Ex527 triggered replication initiation at dormant origins. We found that MTBP was detected on baseline origins with and without Ex527, whereas dormant origins were associated with MTBP only in cells exposed to Ex527. In contrast, pRecQL4 preferentially associated to dormant origins, and this association was markedly reduced when cells were exposed to Ex527 (Supplementary Fig. 1I). These observations suggested that as in cancer cells, the MTBP-TICRR/TRESLIN complex marks activated replication origins in non-transformed cells, whereas pRecQL4 specifically associates with dormant origins.

## The interactions of MTBP-TICRR/TRESLIN with replication origins reflect the order of replication initiation

To determine the timing of MTBP, TICRR/Treslin and pRecQL4 association with replication origins during cell cycle progression, we collected HCT116 cells released from a metaphase block (induced with nocodazole) and followed the progression of MTBP and TICRR/Treslin binding to chromatin (Supplementary Fig. 2A, B). As shown in Supplementary Fig. 2B, association of MTBP and TICRR/Treslin with chromatin was detected at the late G1 stage (2–4 h after release from nocodazole block) and increased during S-phase (6–14 h post release). pRecQL4 was detected on chromatin during S-phase (10–14 h post release). Mapping MTBP and TICRR/Treslin binding sites (Supplementary Fig. 2C, D) demonstrated that during early G1 (3 h post-release), MTBP and TICRR/Treslin showed a low enrichment at the vicinity of replication origins, whereas a strong, distinct colocalization at baseline origins was detected during S-phase (10 h post release). As summarized in Supplementary Fig. 2D, both MTBP and TICRR/Treslin preferentially associated with baseline origins and showed minimal binding to dormant origins and to non-origin sequences.

To characterize the association of pMCM2-S139, MTBP, TICRR/Treslin and pRecQL4 with baseline and dormant origins in early and late replicating chromatin, we utilized a double-thymidine block (DTB), which triggered a synchronous progression through S-phase, in cells with intact SIRT1. DTB synchronized cells at the G1/S boundary (0 h; pre-initiation of DNA replication). Following release from DTB, cells progressed synchronously through early, middle, and late S-phase (1 h, 4 h, and 8 h post-release, respectively; Fig. 2A, B). In parallel, we identified replication origins that initiated replication at distinct times through S-phase by determining the copy-number-based order of DNA replication (replication timing) in asynchronous cells (Supplementary Fig. 2E) and stratified replication origins into early, middle, and late replicating categories. We then concomitantly monitored replication initiation at baseline origins using nascent strand sequencing (NS-Seq; Fig. 2C) and measured the chromatin binding patterns of pMCM-S139, TICRR/Treslin, MTBP, and pRecQL4-S89 (Fig. 2D, E, Supplementary Fig. 2E–G).

We observed that the replication timing stratification mirrored the NS-Seq pattern in cells released from DTB (Fig. 2C). pMCM2-S139 bound to all baseline origins at the onset of S-phase and gradually detached throughout S-phase (Supplementary Fig. 2F). MTBP (Fig. 2D) and TICRR/Treslin (Supplementary Fig. 2G) selectively associated with early replicating origins 1-h post-DTB release, mid-S-phase origins 4 h post-DTB release, and late replicating origins 8 h post-DTB release. In contrast, pRecQL4 showed a minimal colocalization with baseline origins, primarily after the initiation of DNA replication regardless of replication time (Fig. 2E). The association of pRecQL4 with dormant origins increased during the later stages of S-phase (4 h post release from DTB and onwards). These findings suggest that the recruitment of the MTBP-TICRR/TRESLIN complex to baseline origins occurs just before the initiation of DNA replication, marking baseline origins for initiation. pRecQL4 is sequentially recruited to dormant origins during S-phase in concordance with the order of DNA synthesis within their replication timing domains (Fig. 2F). These observations suggest that replication origins that bind pRecQL4-S89 do not associate with the MTBP-Treslin complex, consistent with origin dormancy. Since pRecQL4-S89 was recruited to dormant origins localized at early-replicating domains in mid-S-phase, it is likely that pRecQL4 binding occurred after those dormant origins were passively replicated by replication forks emanating from adjacent baseline origins.

## The MTBP-TICRR/TRESLIN complex dissociates from baseline origins upon replication stress

Replication stress and DNA damage can impede DNA synthesis emanating from baseline origins, leading, upon recovery, to ATR-mediated activation of dormant origins to ensure complete genome duplication[4,6,42]. We have previously shown, consistent with a previous study[43], that ATR is involved in the phosphorylation of MCM2-S108, and demonstrated that this phosphorylation is required for dormant origin initiation[15]. We asked if the association of MTBP-TICRR/TRESLIN

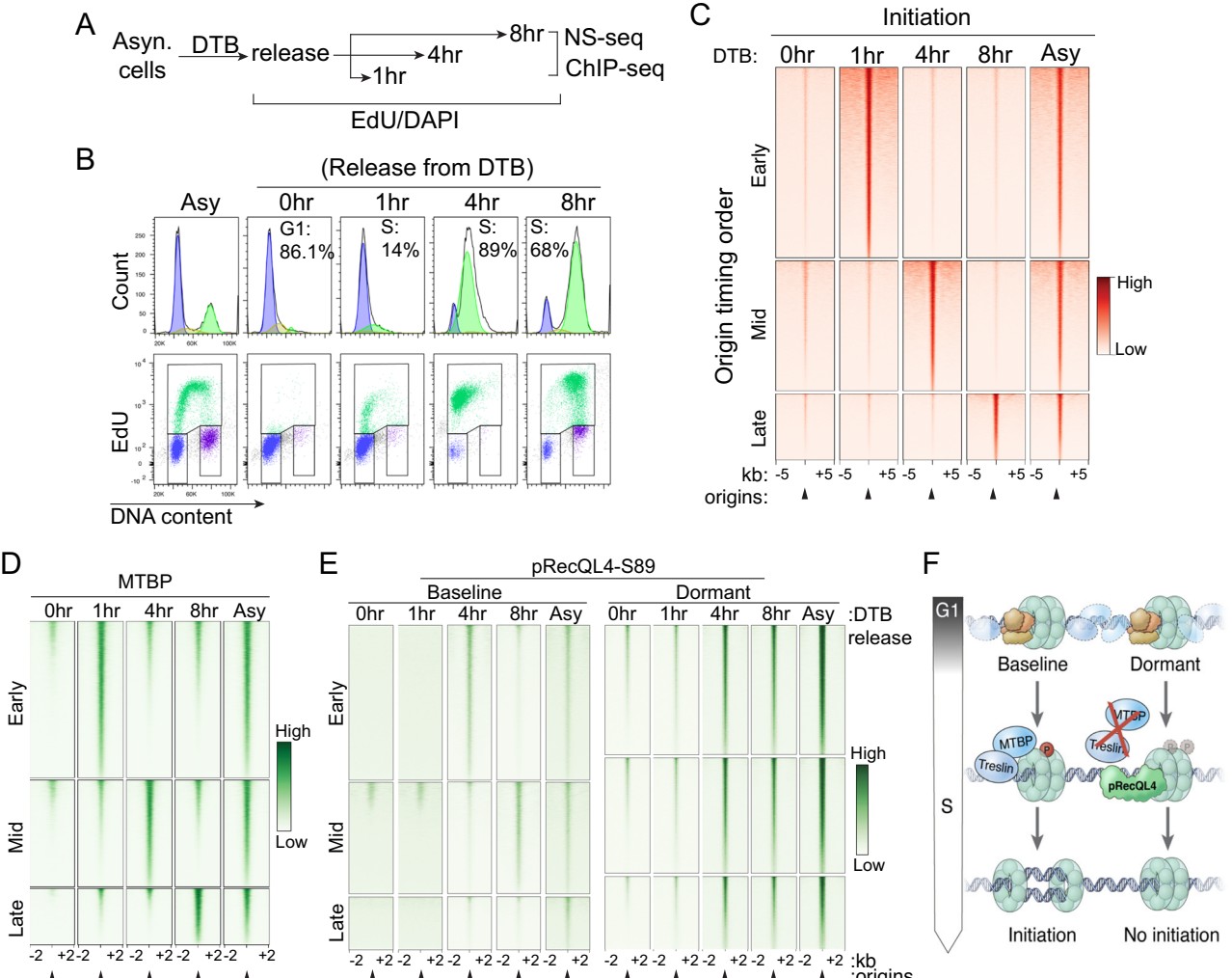

**Fig. 2 | Timing of MTBP and pRecQL4 association with replication origins during S-phase progression.** **A** Experimental procedure. HCT116 cells were released from a double thymidine block (DTB) and collected asynchronously (Asy) or at the indicated time points, corresponding to the early, mid-, and late-S phase of the cell cycle. Replication initiation sites were mapped by NS-seq and chromatin binding patterns of the indicated proteins were determined by ChIP-Seq. **B** Confirmation of cell synchronization by flow cytometry. **C** Nascent strand abundance in HCT116 cells released from DTB, as indicated in panel (**A**), and collected 0 h, 1 h, 4 h, and 8 h post-release and in asynchronous (Asy) HCT116 cells. Early, mid, and late-initiating replication origins were stratified based on replication timing profiles collected as indicated in Supplementary Fig. 2E. ChIP-seq using MTBP (**D**) and pRecQL4-S89 (**E**) antibodies was performed in cells collected at the same time points as in panel (**C**). The heatmaps show MTBP and pRecQL4-S89 signal strengths in early-, mid-, and late-replicating chromatin centered on origins stratified as in panel (**C**). For MTBP, which does not bind dormant origins, only baseline origins are shown. For pRecQL4-S89, the left panel shows binding to baseline origins and the right panel shows binding to dormant origins. **F** A model showing the selective association of MTBP and pRecQL4-S89 with replication origins, regulating the initiation of DNA replication. Dashed ellipses represent diffused binding, whereas solid ellipses indicate enriched binding.

and/or pRecQL4 with dormant origins occurred downstream of the ATR-mediated phosphorylation of MCM2 on S108. As previously reported, we have generated cells harboring a serine-to-alanine mutation at MCM2-S108 (Fig. 3A) by exogenously expressing cDNA for WT-MCM2 or MCM2-S108A in cells which were depleted of endogenous MCM2 by activation of MCM2-mAID with auxin (IAA)[15]. We have previously shown that in these cells, MCM2-S108A cannot be phosphorylated by ATR, and these cells subsequently cannot phosphorylate MCM2-S139 in response to DNA damage or SIRT1 inactivation[15]. Consequently, cells harboring MCM2-S108A exhibit normal replication patterns at baseline origins, but do not initiate replication at dormant origins regardless of SIRT1 status[15]. We utilized the MCM2-S108A mutant to determine chromatin binding patterns of MTBP (Fig. 3B). In cells harboring WT-MCM2, MTBP associated with baseline origins. MTBP also associated with dormant origins after cells were treated with the SIRT1 inhibitor Ex527, which removed SIRT1-mediated suppression of dormant origins. Such association of MTBP

with dormant origins was not observed in cells harboring the MCM2-S108A mutant, regardless of SIRT1 activity (Fig. 3B, left). In parallel, in cells with active SIRT1, dormant origins were associated with pRecQL4-S89 in cells harboring WT-MCM2; however, MTBP, RecQL4, pMCM2-139 and pRecQL4 did not exhibit binding at dormant origins in cells expressing the MCM2-S108A mutant (Fig. 3B, right, Supplementary Fig. 3A). These observations suggest that the association of either pRecQL4-S89 or MTBP with dormant origins requires an intact serine on MCM2 at position 108.

We next examined the dynamics of MTBP and pRecQL4-S89 association with replication origins in cells that encountered replication stress, which has been reported to activate dormant origins[5,6]. To induce replication stress, we exposed cells to 10 µM APH for 1 h and then measured the association of replication origins with MTBP and pRecQL4-S89 during a 36-h recovery period (Fig. 4A, B). The exposure to APH triggered replication stress, significantly reducing the fraction of EdU-positive cells. Normal DNA synthesis levels were gradually

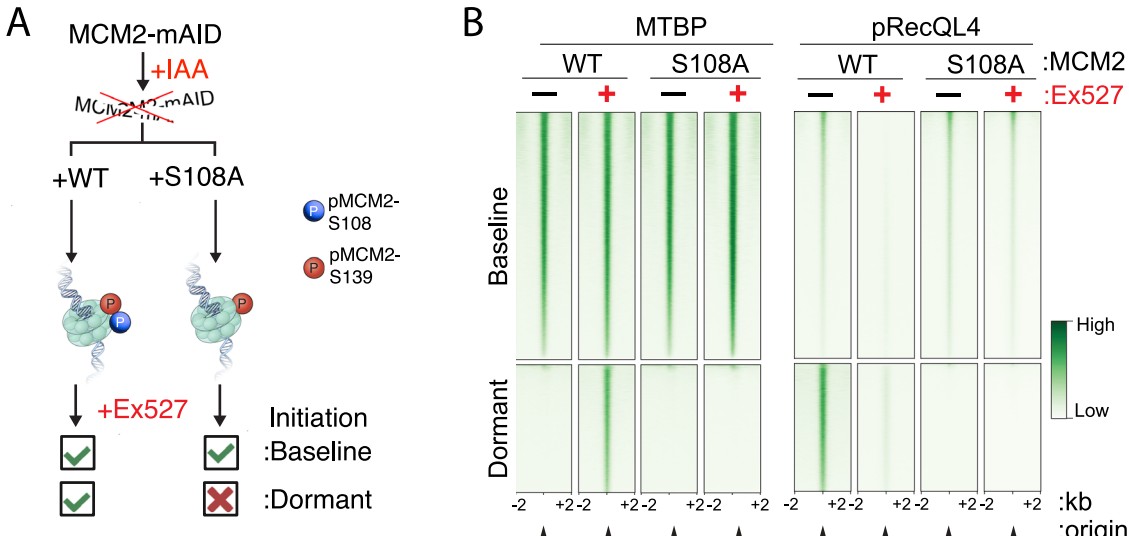

**Fig. 3 | MTBP and pRecQL4 association with dormant origins. A** Effect of the MCM2[S108A] mutation on interactions with MTBP and pRecQL4. Experimental procedure: HCT116 cells with MCM2-mAID were complemented with WT or S108A MCM2 constructs. IAA treatment (500 µM for 16 h) depleted endogenous MCM2-mAID, enabling the incorporation of WT or S108A MCM2 into the pre-replication complex. Treatment with the SIRT1 inhibitor Ex527 (1 mM every 24 h for 2 days) activates dormant origins in cells complemented with the intact, but not the S108A mutated MCM2[15]. **B** Replication origin binding of MTBP (left) and pRecQL4 (right) in unperturbed and Ex527 treated (1 mM every 24 h for 2 days) HCT116 cells harboring either WT-MCM2 or S108A-MCM2. Baseline and dormant origins were categorized as described in the legend to Fig. 1A, B.

restored after APH was washed, reaching normal levels by 36 h after release from APH inhibition. The overall levels of MTBP, RecQL4, and RecQL4-S89 phosphorylation did not change after a 1-h exposure to APH. However, MTBP binding to chromatin was reduced, and concomitantly, pRecQL4-S89 chromatin binding increased (Supplementary Fig. 3B). After removal of APH, MTBP chromatin association increased and pRecQL4-S89 chromatin association decreased, gradually returning to normal levels by 8 h of recovery.

When we mapped the binding sites of MTBP and pRecQL4 to chromatin in unperturbed cells and in cells exposed to APH, we observed that the majority (93%) of baseline origins were associated with MTBP in unperturbed cells, whereas 2.7% colocalized with both MTBP and pRecQL4-S89. Almost no baseline origins were associated solely with pRecQL4-S89 (Fig. 4C, D, Supplementary Fig. 4A,). Exposure to APH reduced MTBP association with baseline origins and increased its association with dormant origins. In addition, we observed some concomitant binding of MTBP and pRecQL4-S89 (Fig. 4C, Supplementary Fig. 4A). When APH was washed out and replaced with APH-free media, baseline origins showed an increased association with pRecQL4-S89, and almost no baseline origins were associated solely with MTBP. For example, an hour after exposure to APH, -68% of baseline origins were associated with pRecQL4-S89 and 28% associated with both pRecQL4-S89 and MTBP (Fig. 4C, D, Supplementary Fig. 4A). After cells were allowed to recover from APH exposure for 8 h (Fig. 4C, D, Supplementary Fig. 4A), about half of baseline origins were solely associated with MTBP, concomitant with a reduced fraction of baseline origins associated with both pRecQL4-S89 and MTBP.

In contrast, 83% of the dormant origins were associated with pRecQL4-S89 in unperturbed cells, whereas 2.1% were associated with both MTBP and pRecQL4-S89 and almost none were associated solely with MTBP (Fig. 4C, D, Supplementary Fig. 4A). After APH exposure and a 1-h release period, -10% of dormant origins were associated with MTBP, 36% with both pRecQL4-S89 and MTBP, and almost none with MTBP alone (Fig. 4C, D, Supplementary Fig. 4A). This association increased from 16% to 51% in 4–8 h after exposure to APH, indicating MTBP's redistribution to dormant origins under replication stress (Fig. 4C, D, Supplementary Fig. 4A). This redistribution coincided with

a reduced fraction of dormant origins associated with both pRecQL4-S89 and MTBP. Recovery from APH exposure for 36 h completely restored both MTBP and pRecLQ4 binding patterns to baseline and dormant origins (Fig. 4C, D, Supplementary Fig. 4A). These observations are consistent with a gradual re-association of MTBP with chromatin concomitant with the loss of pRecQL4-S89 upon recovery from replication stress.

We then asked if origin-bound MTBP and pRecQL4-S89 interacted with each other on chromatin, either during unperturbed growth or after exposure to APH. We first immunodepleted elongating replication forks from chromatin extracts using antibodies against Proliferating Cell Nuclear Antigen (PCNA), a DNA polymerase-associated processivity factor. Then, we used the PCNA-depleted extracts to perform sequential immunoprecipitations with antibodies directed against MCM2, followed by antibodies targeting either MTBP or pRecQL4 (Fig. 4E). We found that pMCM2-S139 was present in both the PCNA-associated and PCNA-depleted chromatin fractions, consistent with the association of the MCM helicase with both pre-ICs and replication forks. In contrast, we did not observe a significant enrichment of either MTBP or pRecQL4 in the PCNA fraction, suggesting that neither protein travels with replication forks. Both MTBP and pRecQL4-S89 associated with pMCM2-S139 in the PCNA-depleted fraction, implying that these two proteins bind pre-RCs, but we did not detect an association between MTBP and pRecQL4-S89 unless cells were exposed to APH.

These observations suggest that MTBP and pRecQL4-S89 form separate, distinct complexes. Similarly, Proximity Ligation Analysis of the colocalization between pRecQL4-S89 and MTBP demonstrated minimal interactions between these two proteins in cells undergoing DNA synthesis (EdU positive; Supplementary Fig. 4B, C) whereas each of the two proteins was associated with pMCM2-S139.

**Phosphorylated RecQL4 is required for replication stress-induced MTBP chromatin dissociation and subsequent recovery**
We next asked if RecQL4 phosphorylation directly impacted MTBP dissociation from baseline origins during APH-induced replication stress. We mapped the replication initiation (NS-seq) and the distribution of MTBP with and without exposure to APH in cells harboring

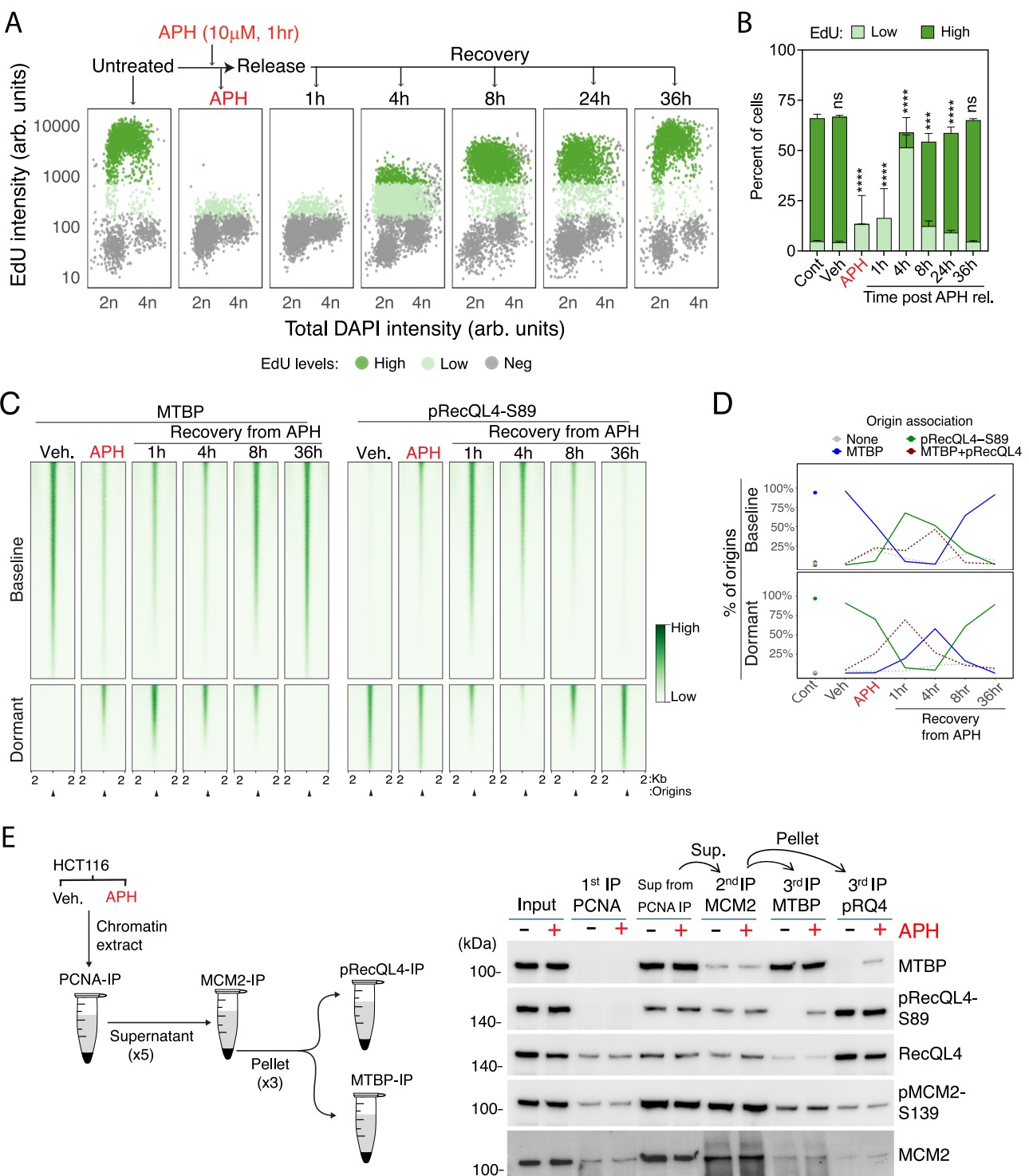

WT-RecQL4, RecQL4-depleted, or an engineered phosphorylation-deficient mutant RecQL4-S89A (Fig. 5A). During normal cell proliferation, MTBP was bound to dormant origins in RecQL4-deficient cells or in cells expressing the phosphorylation-deficient RecQL4 mutant. This binding was not observed in cells with intact RecQL4 (Fig. 5A, right), mirroring the active origins identified by NS-seq (Fig. 5A, left). Intriguingly, cells containing the RecQL4-S89A mutant exhibited proliferation rates and cell cycle distributions similar to cells derived from the same RecQL4-depleted parental cell line harboring the unmodified RecQL4 (Supplementary Fig. 5A, B). However, exposure to APH

induced MTBP dissociation from 71% of baseline origins (Fig. 5B, Supplementary Fig. 5C, D) in cells containing intact RecQL4, whereas MTBP did not dissociate from baseline origins in RecQL4-deficient cells (Fig. 5B, middle) or in cells harboring the phospho-deficient mutant RecQL4-S89A (Fig. 5B, right, see Supplementary Fig. 5C, D for quantification). These observations suggest that RecQL4 phosphorylation at S89 is required for the removal of MTBP from the pre-replication complexes upon replication stress.

We then asked if pS89-RecQL4-regulated MTBP dissociation from chromatin was required for efficient recovery from replication stress.

**Fig. 4 | MTBP and pRecQL4 association with baseline origins after perturbation of DNA replication. A** Recovery of HCT116 cells from APH-induced replication stress. Top, experimental procedure. Below, representative quantitative immunofluorescence-based cytometry (QIBC) of cells exposed to APH (10 μM for 1 h) and released for the indicated time frames. **B** EdU incorporation in cells treated as indicated in panel **A**, quantified across 4 biological replicates: Control (*n* = 5010), APH (*n* = 5706), 1 h (*n* = 5648), 4 h (*n* = 11,951), 8 h (*n* = 8528), 24 h (*n* = 7498), and 36 h (*n* = 5024). Stacked bars show the mean, error bars indicate standard deviation (SD). Statistical analysis was performed for the "high EdU" population compared to the control using a two-sided paired Tukey test, adjusted for multiple comparisons: ***$p$ = 0.0008; ****$p$ < 0.0001; ns not significant. **C** Binding of MTBP (left) and pRecQL4-S89 (right) to baseline and dormant origins in HCT116 cells exposed to aphidicolin (APH, 10 μM for 1 h) and released at the indicated time points.

**D** Fractions of baseline (top) and dormant (bottom) origins binding to MTBP, pRecQL4, or both, at the indicated timepoint post-APH removal. Vehicle (DMSO) treated (veh.) and untreated control (cont.). The line plot represents the average of two biological replicates. **E** Replication stress facilitates an interaction between MTBP and pRecQL4. Left, experimental strategy. Right, Chromatin extracts from asynchronous cells untreated or treated with APH (10 μM for 1 h) were first immunoprecipitated (IP) with PCNA. PCNA-depleted fractions were then immunoprecipitated (pooled five reactions) with pMCM2 antibodies to isolate pre-replication complexes. MTBP or pRecQL4 antibodies were used for further IPs (with three pooled reactions), and the presence or absence of MTBP or pRecQL4 was detected by immunoblotting. Representative immunoblots of 3 independent replicates. Source data are provided as a Source Data file.

Quantitative imaging-based cytometry (QIBC) revealed that APH treatment halted replication in cells harboring intact and mutated RecQL4 alike, but cells expressing the RecQL4-S89A mutant showed a reduced proportion of DNA synthesis (EdU incorporation) and increased retention of RPA2 on chromatin upon recovery (Fig. 5C, D, and Supplementary Fig. 5E). These findings, which indicate the presence of single-stranded DNA at under-replicated sites, suggest that DNA replication was stalled in these cells, implying an inefficient recovery. In concordance, the mutant RecQL4-S89A also showed a marked accumulation of the DNA damage marker γH2AX upon recovery from APH (Fig. 5E, Supplementary Fig. 5F, G).

To assess the impact of inefficient recovery in RecQL4-S89A cells on viability, we next checked the long-term survival of cells harboring WT-RecQL4 and S89A-RecQL4 after a temporary exposure to APH (Fig. 5F). In cells harboring RecQL4-S89A, 84% and 92% of cells did not survive following a 6 or 8-hour-long exposure to APH, respectively, whereas this exposure had a minimal effect on the survival of cells harboring intact RecQL4.

RecQL4 was implicated in genomic stability, and in particular, in the resolution of telomeric D-loops in concordance with the faulty telomeric replication exhibited by ReQL4 mutants[35,47,48]. As RecQL4 binds G-quadruplexes, which are associated with replication origins as well as telomeric repeats, we asked if the phosphorylation of ser 89 also plays a role in telomere stability by utilizing TelSeq to measure telomere length in cells harboring the RecQL4-S89A mutant. We observed that the RecQL4-S89A mutant exhibited decreased telomere length in both ALT-positive and ALT-negative cells, suggesting telomere erosion independent of ALT activation or telomerase activity (Supplementary Fig. 5H). These findings underscore the critical role of pRecQL4-S89 in facilitating efficient recovery from replication stress and maintaining genomic stability, potentially impacting cell viability under conditions of genomic challenge.

## Discussion

The studies reported here suggest that baseline and dormant origins differ in their binding patterns for replication initiation factors, downstream of replication licensing. During unperturbed growth, baseline origins recruit the MTBP-TICRR/TRESLIN complex prior to the initiation of DNA replication, whereas dormant origins associate with pRecQL4-S89. When replication is perturbed, the MTBP-TICRR/TRESLIN complex dissociates from baseline origins. As cells recover from replication stress, MTBP-TICRR/TRESLIN is redistributed to associate with both dormant and baseline origins. MTBP-TICRR/TRESLIN redistribution, and the subsequent recovery from replication stress, require RecQL4-S89 phosphorylation. Our observations are consistent with a model whereby pRecQL4-S89-dependent dissociation of MTBP-TICRR/TRESLIN from baseline origins, and its subsequent redistribution to dormant origins, facilitates initiation from those origins and permits the completion of DNA replication upon recovery from replication stress (Fig. 6).

Replication origin selection is a critical step in cell proliferation that imposes a distinct replication order (also known as "replication timing") to coordinate replication and transcription and prevent fragility caused by transcription-replication collisions[2,3,6,49]. The assembly of pre-RCs on origins at the G1 phase (replication licensing) is temporally and spatially constrained to avoid re-replication. Because mammalian pre-RCs do not exhibit sequence-specific DNA binding[2,6,49] and associate with baseline and dormant origins alike, pre-RC binding by itself cannot determine whether replication would initiate at particular locations. The studies reported here suggest that replication origin choice occurs during the S-phase via specific interactions with the MTBP-TICRR/TRESLIN complex. These findings align with previous studies indicating that the MTBP-TICRR/TRESLIN complex associates with transcription start sites[17,24], which often colocalize with replication origins[49,50]. Because MTBP-TICRR/TRESLIN associates with origins just prior to initiation, MTBP-TICRR/TRESLIN binding can serve as a marker for active replication initiation sites. The use of MTBP or TICRR/Treslin binding as a surrogate for the direct mapping of replication initiation events can facilitate the study of replication dynamics, especially in samples that currently are not amenable to replication origin mapping due to sample size and technical limitations.

Our data suggest that the MTBP-TICRR/TRESLIN complex associates with origin-assembled pre-RCs during S-phase, just prior to the initiation of DNA replication. These observations seemingly differ from previous studies suggesting that the MTBP-TICRR/TRESLIN complex associates with chromatin during the G1 phase of the cell cycle[20,22]. In addition, previous observations suggested that MTBP associates only with replication initiation sites in early-replicating chromatin in asynchronous cells[51], whereas we detected MTBP at all origins, including late-replicating ones, albeit at different times during S-phase. These apparently contradictory observations can be reconciled by our finding that the MTBP-TICRR/TRESLIN complex exhibits low, diffuse chromatin binding during the G1 phase, and that this binding increases and localizes to replication origins during the S-phase. In addition, we found that MTBP-TICRR/TRESLIN's association with baseline origins prior to initiation is temporary, consistent with the proposal[20] that MTBP and TICRR/Treslin exhibit non-selective chromatin association during G1, which is subsequently restricted to pre-RCs at replication origins following the increase in DDK activity. The observed selectivity of MTBP binding sites at early origins in asynchronous cells could reflect the higher abundance of origins at early-replicating chromatin. The selective association of MTBP-TICRR/TRESLIN with pre-RCs just prior to origin activation suggests that MTBP-TICRR/TRESLIN binding could serve as a dynamic marker for origin activity, aiding in the assessment of the DNA damage response and dormant origin activation in non-transformed cells and limited-size samples.

The origin selection process creates an excess of potential initiation sites, allowing for flexibility in the replication initiation process by licensing backup initiation sites. This is evidenced by the replication stress and chromosomal fragility that occur when replication licensing

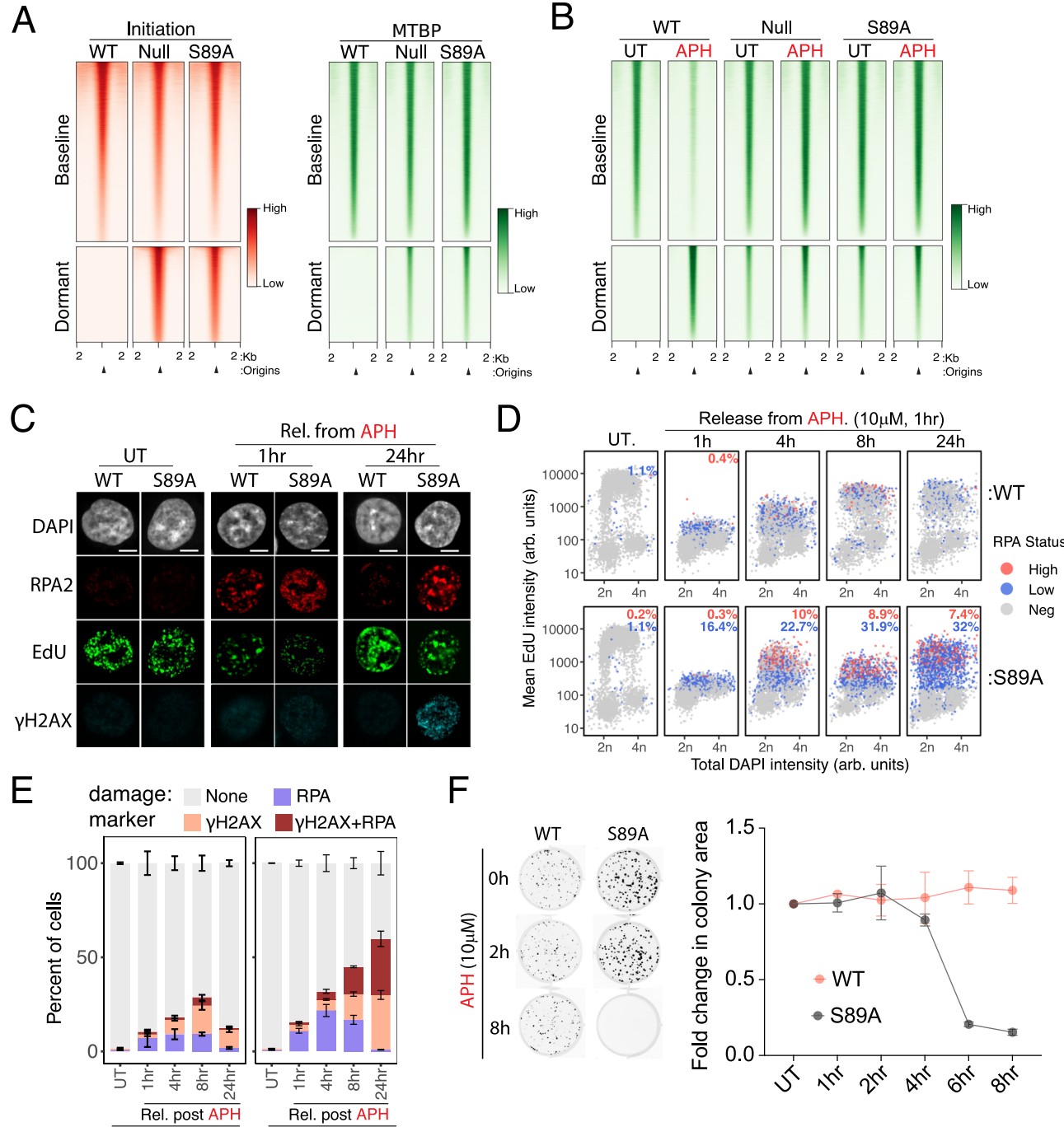

**Fig. 5 | Role of RecQL4-S89 phosphorylation in regulating MTBP association with replication origins. A** Nascent strand abundance (left, red) and MTBP association (right, green) at baseline and dormant replication origins in unperturbed RecQL4 CRISPR knockout HCT116 cells (Null) complemented with either WT-RecQL4 or S89A-RecQL4. Baseline and dormant origins were stratified as described in the legend to Fig. 1. **B** MTBP binding at baseline and dormant replication origins in the cell variants described in panel A, without APH or 1 h after release from APH exposure (10 μM for 1 h). **C, D** The recovery of HCT116 cells from exposure to APH. Representative immunofluorescence (**C**) and Quantitative immunofluorescence-based cytometry quantification (**D**) of DAPI (a DNA content indicator), EdU (a DNA synthesis marker), RPA2 (a replication stress marker), and γH2AX (a DNA damage marker). HCT116 cells harboring either WT-RecQL4 (WT) (**D**, top panel) or S89A-RecQL4 (**D**, bottom panel) were untreated (UT) or exposed to APH (10 μM for 1 h) and then released (rel.) into a fresh medium for the indicated time periods. For each

condition, cells were pre-labeled with EdU (10 μM for 30 min) and then detergent-extracted and fixed (see methods). Scale bar = 2 μm. EdU versus DAPI levels were plotted, and high-, low-RPA positive and RPA negative (neg.) cells (for calibration, see Supplementary Fig. 4E) are shown in red and blue, respectively. A minimum of 4 replicates were analyzed and plotted together. **E** Quantification of RPA and γH2AX in untreated cells and in cells recovering from exposure to APH as described in Fig. 5C, D. (for QIBC profiles, see Supplementary Fig. 5G and source Data file). Total number of cells quantified across 4 biological replicates: for WT- UT (*n* = 5010), 1 h (*n* = 5648), 4 h (*n* = 11,951), 8 h (*n* = 8528), 24 h (*n* = 7498) and S89A-UT (*n* = 5024), 1 h (*n* = 6443), 4 h (*n* = 8601), 8 h (*n* = 9288), 24 h (*n* = 6186). Stacked bars show the mean, error bars indicate SD. **F** Left, Colony formation in HCT116 cells harboring WT (left) and S89A-RecQL4 (roight) exposed to 10 μM APH for the indicated times. Right, mean colony formation in three independent experiments. Error bars indicate SD. Source data are provided as a Source data file.

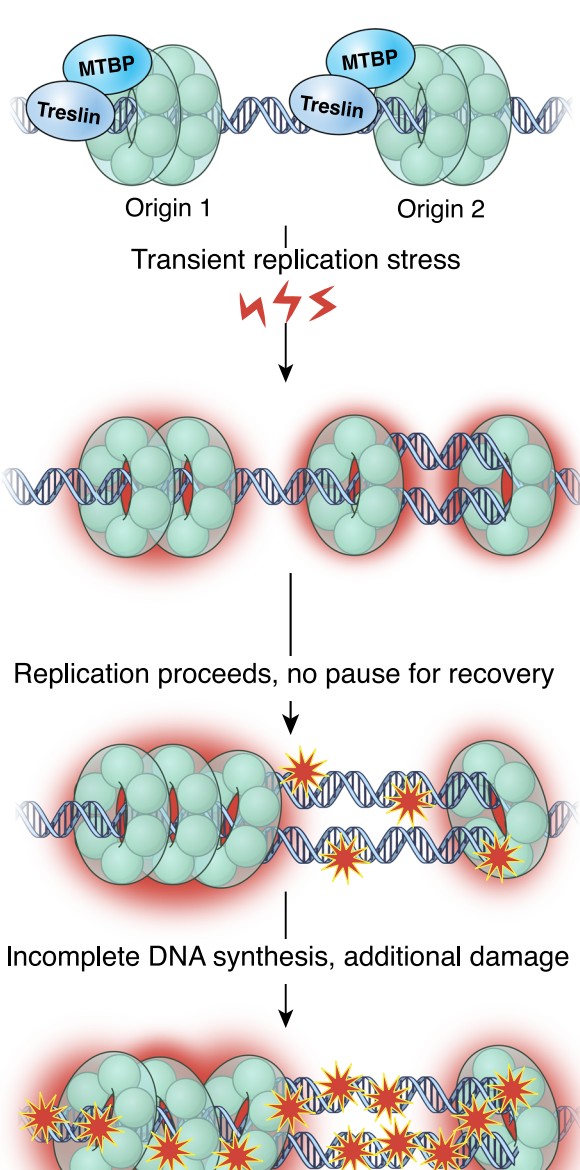

**Fig. 6 | A model illustrating dynamic interactions at replication origins during the recovery from replication stress.** Left, in cells harboring WT-RecQL4, recruitment of the Treslin-MTBP complex marks a sub-group of baseline origins (e.g. origin 1) whereas pRecQL4-S89 (pRecQL4) associates with dormant origins (e.g. origin 2). When cells encounter replication stress, baseline origins can stall (red halo) but dormant origins, which do not initiate replication, are not affected and still maintain MCM complexes (green ring). Although pRecQL4 binding does not allow initiation from dormant origins under normal circumstances (see Fig. 2F), RecQL4 dissociates from origins when cells recover from replication stress and allows MTBP re-association with those origins. Upon the binding of MTBP and Treslin to dormant origins, these origins initiate replication to complete DNA synthesis. Right, in cells that do not harbor pRecQL4-S89 (either RecQL4 depleted or harboring the RecQL4-S89A substitution), MTBP and Treslin associate with both origins 1 and 2, and replication initiates from both origins as cells encounter acute replication stress. Under these conditions, replication from dormant origins cannot rescue the damage after replication stress is removed, preventing normal recovery and leading to the accumulation of ssDNA and subsequent DNA damage.

is reduced due to mutations in pre-RC components or at genomic areas with sparse coverage of licensed replication origins such as common fragile sites[5,52]. However, excess initiation at licensed origins is also deleterious, instigating R-loops and chromosomal rearrangements that impede genomic stability[4,44,52,53]. Our observations suggest that the MTBP-TICRR/TRESLIN complex, along with the phosphorylated form of RecQL4-S89, plays an essential role in the determination of origin activation and silencing to modulate the frequency of replication initiation. Mechanistically, we find that the phosphorylation of

RecQL4-S89 and its recruitment to chromatin at dormant origins prevent the recruitment of MTBP-TICRR/TRESLIN, and prohibit replication initiation at those origins during unperturbed cell proliferation. Notably, although pRecQL4-S89 binding to chromatin requires the assembly of pre-RCs (e.g. binding does not occur in cells with activated MCM2-mAID), the association of pRecQL4-S89 might occur after dormant origins are duplicated passively by replication forks emanating from adjacent baseline origins, prohibiting further initiation and maintaining origin dormancy. When cells encounter replication

stress, MTBP-TICRR/TRESLIN dissociates from baseline origins in a pRecQL4-dependent manner, as is evident by the fact that MTBP is retained on baseline origins in the RecQL4-S89A mutant. In cells that contain intact RecQL4, MTBP-TICRR/TRESLIN eventually redistributes to associate with both baseline and dormant origins along with the activation of replication initiation at dormant origins.

When cells are exposed to APH, ATR phosphorylates MCM2-S108, which in turn allows MCM2-S139 phosphorylation at dormant origins[15,43]. MCM2-S139 phosphorylation is a prerequisite for MTBP binding to dormant origins. We have previously reported that ATR-mediated phosphorylation of MCM2-S108 is essential for the activation of dormant origins in cells undergoing replication stress[15]. Our current findings further indicate that phosphorylation of MCM2-S108 is required for MTBP redistribution to dormant origins, and that neither MTBP nor pRecQL4 bind dormant origins when MCM2-S108 is substituted with an alanine residue that cannot be phosphorylated. The requirement for an intact MCM2-S108 for MTBP and pRecQL4 binding at dormant origins is not alleviated by the inactivation of SIRT1. Taken together, these observations suggest that the presence of serine at position 108 allows pRecQL4 and/or Treslin-MTBP to recognize and associate with dormant origins. In cells with intact MCM2-S108, dormant origins preferentially associate with pRecQL4, and this phosphorylation prevents Treslin-MTBP binding and subsequent initiation at dormant origins during unperturbed growth.

pRecQL4-mediated dormant origin suppression is alleviated upon replication stress, as ATR-mediated MCM2-S108 phosphorylation facilitates DDK-mediated phosphorylation at MCM2-S139[15], concomitant with the dissociation of pRecQL4 and the redistribution of MTBP-TICRR/Treslin to bind both baseline and dormant origins. MTBP-TICRR/Treslin redistribution to dormant origins is critical for the recovery of cells from replication stress, as is evident from the low viability of cells harboring the mutated RecQL4-S89A after a short exposure to APH. However, our observations indicate that removal of pRecQL4-S89 suppression is not sufficient to activate dormant origins, as dormant origins did not initiate replication in cells harboring the MCM2-S108A mutant despite the absence of pRecQL4-S89 binding. Taken together, our observations are in line with a scenario whereby the assembly of pre-RCs harboring intact MCM2-S108 at dormant origins is required for subsequent pRecQL4-S89 binding, which prevents MTBP and TICRR/Treslin recruitment and consequently suppresses initiation at those origins unless MCM2-S108 is activated by ATR. In cells with intact MCM2-S108, if pRecQL4-S89 is not present to interact with dormant origins (for example, due to a deletion or a mutation on serine 89), MTBP and TICRR/Treslin can bind those origins regardless of ATR activity, initiating excess replication.

Because MTBP, TICRR/Treslin and RecQL4 interact with members of the pre-RC and the replisome, it is tempting to assume a potential regulatory mechanism whereby the MTBP-TICRR/TRESLIN/pRecQL4-S89 exchange toggles between origin dormancy and activation. Such a mechanism would link the DNA damage response to DNA repair pathways and is consistent with an inhibitory role of RecQL4 in the initiation of DNA replication, potentially via its DONSON-like SLD2 domain[37,38]. However, our observations cannot rule out an indirect role of pRecQL4-S89 via the activation of DNA repair interactions, which prevent DNA damage sensing and affect MTBP-TICRR/TRESLIN chromatin association. We also cannot rule out a mechanism whereby the phosphorylation of chromatin-bound RecQL4, not the recruitment of pRecQL4-S89, modulates MTBP-TICRR/TRESLIN binding. Of note, IR-induced CDK1/2-mediated phosphorylation of RecQL4 at S89 and S251 regulates DNA repair pathway choice, affecting RecQL4 recruitment to damage sites and DNA unwinding activity as well as its interaction with the replisome component MCM10[29,54]. As SIRT1, which prevents binding of TOPBP1 to DNA, deacetylates RecQL4, this deacetylation can play a role in DNA damage signaling[46].

TICRR/Treslin and MTBP are CDK1/2 substrates that also associate with TOPBP1, an ATR kinase partner and a SIRT1 substrate[15,39,40]. Hence, TICRR/Treslin-MTBP-TOPBP1 interactions provide a potential convergence point between the replication initiation machinery and the DNA damage response[20,55,56]. For example, one might speculate that selective CDK1/2 mediated phosphorylation can either facilitate pre-IC assembly, by phosphorylating MTBP-TICRR/TRESLIN, or disallow this assembly by phosphorylating RecQL4, thus preventing it from acting downstream of DONSON to facilitate# the conversion of the two MCM complexes into separate replication forks. Such a mechanism could be in line with the observed enrichment of RecQL4 at replication origins and G-quadruplex sites as well as the positive and negative functions reported for RecQL4 in the initiation of DNA replication[25,27−29,34,36−38]. Notably, RecQL4 is also a SIRT1 substrate, and SIRT1-mediated deacetylation of RecQL4 plays a role in the repair of oxidative base damage[46]. Our studies do not rule out a direct role of SIRT1 in modulating the function of RecQL4 in DNA replication. Finally, RecQL4 is involved in pathways for DNA damage signaling and DNA repair, and our observations suggest that it also plays a central role in containing damage from replication stress by preventing over-activation of replication origins during unperturbed growth as well as recovery from temporary attenuation of DNA replication.

Excessive activation of replication origins can lead to genomic instability, making pathways that prevent excessive replication or eliminate surplus DNA targets for therapeutic intervention. By understanding the signals that trigger origin activation and alter post-licensing origin choice in the face of replication stress, therapeutic advances can be made. Our study suggests that indeed there is a separate signaling pathway that regulates dormant origins. We found that this pathway requires RecQL4, a protein facilitating DNA repair and mitochondrial maintenance whose disruption can lead to several devastating human diseases associated with premature aging and cancer predisposition. Beyond providing insights into the role of RecQL4 as a possible link between metabolic activity and genome maintenance, this finding opens possibilities for further investigations to elucidate how dormant origins allow cancer cell proliferation in the wake of persistent replication stress and whether these signals can be targeted for cancer treatment.

## Methods

### Cell culture, chemicals, and establishment of stable cell lines

HCT116 (CCL247) and U2OS (HTB96) cell lines, both are from ATCC. Fibroblasts lines AG04446 collected from a healthy individual donor at age 48 were obtained from the Coriell Institute's Aging Cell Depository. The SIRT1[Null], WT[SIRT1] and Mut[SIRT1] were reported earlier[15,41]. Briefly, the SIRT1 gene was deleted from HCT116 cells using a CRISPR-Cas9 vector targeting exon1, and clones were confirmed using Sanger sequencing. WT-SIRT1 cDNA was mutated at H363Y using a site-directed mutagenesis kit (NEB, cat# E0554S). The RecQL4 gene was depleted from HCT116 cells using a CRISPR-Cas9 kit (Santa Cruz, cat # sc-403224), and clones were confirmed using western blotting for RecQL4. WT-RecQL4 cDNA was mutated at S89A using a site-directed mutagenesis kit (NEB, cat# E0554S). HCT116 cells harboring MCM2-mAID (endogenous MCM2 tagged with auxin inducible degron) expressing WT-MCM2 or phospho-deficient mutant MCM2 for S139A or S108A were reported earlier[15]. HCT116 cells and the stable clones generated in HCT116 were grown at 37 °C in a 5% CO2 atmosphere in RPMI medium (Thermo Fisher, cat# 11875-119), supplemented with 10% FBS (GeminiBio, cat# 100-106-500). All original cancer cell lines were obtained from ATCC, and all cell lines tested negative for mycoplasmas (Lonza, LT07−418). Fibroblast lines AG04446 collected from a healthy individual donor at age 48 were obtained from the Coriell Institute's Aging Cell Depository. Fibroblasts were immortalized by human Telomerase Reverse Transcriptase protein (hTERT) stable expression

using the hTERT Cell Immortalization Kit (ALSTEM, cat# CILV02) as per the manufacturer's instructions and grown at 37 °C in a 5% CO2 atmosphere in MEM (Thermo Fisher, cat# 11095080) medium with 15% FBS. Ex527 (cat# E7034), Aphidicolin (cat# A0781), indol-acetic acid (IAA, auxin for MCM2-mAID degradation) (cat#I5148), nocodazole (cat# M1404) and thymidine (cat# T3763) were purchased from SIGMA.

## Colony assay

Cell survival was assessed by colony formation assay. Two thousand cells were plated on 6 well plate. Thirty-six hours after seeding, cells were incubated with 10 μM aphidicolin for 1, 2, 4, 6 and 8 h then washed twice with warm medium. After 10 days of incubation, colonies were fixed with methanol for 10 min, then stained with a solution of 0.1% crystal violet (SIGMA, cat# C0775) in water for 16 h. Colonies were counted using ImageJ (https://imagej.net/ij/). Clonogenic survival curves were constructed from triplicates.

## Flow cytometry

Cells were pulse-labeled with 10 μM EdU for 30 min before harvest. EdU staining was performed using the Click-iT EdU kit (ThermoFisher, cat# C10634 (for AL647) or cat# C10633 (AL488)) according to the manufacturer's protocol. A BD LSR Fortessa cell analyzer with FACS-Diva software and/or FlowJo10.6 was used for cell cycle analysis. See source data for the gating strategy.

## Antibodies

The primary antibodies used were RecQL4 (Proteintech # 17008-1-AP and Cell Signaling, cat# 2814), MCM2 (Cell Signaling, cat# 12079), phospho-MCM2 S139 (Cell Signaling, cat# 12958), MTBP (Novus biologicals, cat# NBP1-86408), PCNA (Millipore cat# MAB424R) RPA2 (Millipore cat# MABE285), Treslin (ThermoFisher cat# PA583839), gamma-H2AX (Millipore cat# 05-636) and histone H3 (Millipore, cat# 07-690). Antibodies against pSIRT1 T530[41] and pRecQL4 S89 (this work, see fig S1C for validation) were custom-made. Anti-Rb and anti-MS HRP-labeled antibodies (Cell Signaling, cat# 7074 and cat# 7076) or AL488 or AL647 (ThermoFisher cat# A11008, A21235) were used as secondary antibodies for western blotting.

## Cell synchronization

For early-G1 cell synchronization, HCT116 with endogenously tagged MCM2-mAID cells harboring WT-MCM2 and S139A-MCM2 were synchronized in mitosis by a shake-off after 16 h of incubation in 100 nM nocodazole. Mitotic cells were washed three times in warm RPMI media without FBS and either collected immediately (0 h) or released in fresh medium for the indicated time periods.

For G1/S and S phase synchronization, HCT116 cells were synchronized at various cell cycle stages using a double thymidine block (DTB). Exponentially growing HCT116 cells at 50–60% confluence were treated with thymidine to a final concentration of 2.5 mM and incubated for 18 h to arrest them at the G1/S boundary. After incubation, cells were washed twice with warm media to remove thymidine and a fresh medium was added. The cells were then allowed to recover for 9 h. Subsequently, thymidine was added again to a final concentration of 2 mM and the cells were incubated for another 17 h to block them at the G1/S boundary for the second time. The cells were washed twice with warm media to remove thymidine and fresh medium was added to release the cells from the block. Samples were collected at desired time points post-release for downstream analyses, such as cell cycle progression studies or DNA replication analyses.

## Chromatin fractionation and immunoblotting

For whole cell lysates, $1 \times 10^6$ harvested cells were washed with 1xPBS and protein were extracted using standard Laemmli buffer. For measurements of chromatin binding, $5 \times 10^6$ harvested cells were incubated in cytosolic extraction buffer (10 μM HEPES, pH 7.9; 10 μM KCl; 1 μM EGTA; 0.25% NP40; 1X protease inhibitor cocktail and phosphatase inhibitor cocktail) for 10 min on ice. Nuclei were collected using centrifugation at $2700 \times g$ for 5 min at 4 °C and washed twice with a cytosolic extraction buffer (without NP40) and resuspended in cold nuclear extraction buffer (20 mM HEPES [pH 7.9], 420 mM NaCl, 20% [v/v] glycerol, 1 mM EDTA, 1 μg/ml RNase A and 20 U/ml DNase, 1× proteinase inhibitor cocktail and phosphatase inhibitor cocktail) and incubated on ice for 10 min and chromatin fractions were collected by centrifugation at $5000 \times g$ for 5 min at 4 °C. DNA remnants from chromatin fractions were then digested using benzonase (20U/100 ml) at 20 °C for 10 min. For immunoblots, chromatin-bound fraction lysates were mixed with 200 μl 1 × SDS loading buffer per 1 million cells. Samples were heated at 100 °C for 5 min, centrifuged and the supernatants used for immunoblots.

## Sequential re-immunoprecipitation

Chromatin-bound fractions were collected from $1 \times 10^7$ cells as described above, and the supernatant adjusted to 200 mM NaCl, 50 mM Tris-HCl pH 7.4, 0.05% Tween-20.

For the first immunoprecipitation (PCNA depletion), soluble chromatin samples were precleaned with Protein G agarose beads (Pierce cat# 20397) for 1 h at 4 °C, then incubated with PCNA antibody at 4 °C overnight. Supernatants were collected with centrifugation at $1500 \times g$. Supernatants were re-immunoprecipitated by re-incubation with PCNA antibody at 4 °C for 3 h and collected using centrifugation to maximize PCNA depletion. We pooled 5 of such reactions in order to obtain enough material to perform sequential co-IP. PCNA-depleted supernatants were then subjected incubated with MCM2 antibody at 4 °C overnight and immunoprecipitations were performed using protein G agarose beads. The immunoprecipitated complex associated with protein G bead was collected using centrifugation at $1500 \times g$. The immunoprecipitated complex was then eluted from beads with 100 ul of 1 M Glycine solution at pH = 2.5, at room temperature for 15 min with gentle agitation. Eluates from 3 such reactions were pooled and then pH adjusted by diluting with 800 ml of 50 mM Tris-HCl pH 7.4. The diluted eluates were then re-immunoprecipitated with either MTBP or pRecQL4 antibodies. For pRecQL4-IP, precipHen immunoprecipitation reagent (Aves labs cat# P1010) was used instead of protein G beads. All bead–antibody complexes were washed three times with TBST (Tris buffered saline, .05% Tween-20. pH 7.5) and resuspended in SDS-PAGE loading buffer. After heating for 10 min at 90 °C, the proteins were analyzed by Western blotting according to standard procedures.

## Chromatin immunoprecipitation and sequencing (ChIP-seq)

Cells were crosslinked with 1% formaldehyde for 10 min at room temperature, and the remaining formaldehyde was quenched with glycine (1.25 mM) and washed twice with PBS. Cell nuclei were isolated using cytoplasmic extraction buffer with 0.25% NP40 buffer (volume of 5 times the pellet size or 500 μl, whichever was higher) plus protease and phosphatase inhibitors (1x) and incubated on ice for 5 min. Nuclei were collected by centrifugation at $2700 \times g$ and resuspended in 500 μl NP40 buffer followed by sonication (40% aptitude, 1 s pulse, 65–80 pulses). The supernatants (about $2 \times 10^6$ cells) were precleared with protein G beads and incubated with specific antibodies and 80 μl of protein G beads overnight at 4 °C. For pMCM2-S139, cells were synchronized in G1/S using a double thymidine block (DTB) and processed as described above. For pRecQL4 ChIP, precipHen immunoprecipitation reagent (Aves labs) was used instead of protein G beads. Beads were washed twice with each of the following buffers: low salt buffer, high salt buffer, lithium chloride buffer, and TE (each spin at $1000 \times g$ for 1–2 min). Samples were eluted, incubated at 65 °C overnight for reverse-crosslinking, then purified by a Monarch PCR & DNA Cleanup Kit (NEB cat# T1030S).

## Nascent strand DNA sequencing (NS-seq)

HCT116 cells and immortalized fibroblasts without or with indicated treatments were harvested, genomic DNA was purified, and nascent strands were isolated as described previously[15] Briefly, DNA was denatured by boiling for 10 min, immediately cooled on ice, and fractionated on a neutral sucrose gradient. Fragments of 0.5–2 kb (containing nascent strand DNA and broken genomic DNA) were collected and treated with λ exonuclease to remove non-RNA-primed broken genomic DNA. The remaining single-stranded nascent strand DNA were converted to double-stranded DNA using the BioPrime DNA Labeling System (ThermoFisher, cat# 18094011). Double-stranded nascent DNA (1 μg) was sequenced using the Illumina genome analyzer II (Solexa). RNAseA pretreated, λ-exonuclease digested controls were used for peak calling as described[15].

## Nascent strand and ChIP-sequencing analysis

Sequencing analyses were conducted following established protocols. Raw FASTQ sequencing files underwent initial trimming with Trimmomatic (version 0.36) and Trim Galore (version 0.4.5) to remove low-quality reads. Quality checks were performed using FastQC (version 0.11.5). Trimmed reads were aligned to the hg19 genome using the BWA aligner (version 0.7.17) and then normalized to each other using BAMscale5 '−scale smallest' to generate normalized bigwig files. Peak calling for ChIP-seq was carried out using the MACS2 (version 2.1.1.20160309) method, with no-IP control sequencing used as controls. For nascent strand sequencing, peaks were called against pre-RNAse treated, exonuclease digested nascent-strand DNA. Dormant origin locations were determined by subtracting the baseline origin peaks detected in WT[SIRT1] containing cells from the total origin peaks detected in Mut[SIRT1] containing cells, using data from both replicates. We created averaged bigwig files using deeptools' bigwigAverage function to visualize genomic coverage and generating heatmaps. Heatmaps were made by first generating a matrix with computeMatrix for each sample bigwig file (averaged bigwigs of two independent replicates) and using baseline and dormant origins as bed coordinates (stratified in Fig. 1A), then plotting with plotHeatmap. Normalized sequencing coverage for NS-seq of ChIP-seq were calculated using deeptools' multiBigwigSummary BED-file function and representative line plots, stacked bar plots and XY-scatter plots were generated in R using ggplot2. Venn diagrams and origin comparison tables were generated by analyzing replicated peaks (common to both replicates) under the indicated conditions using bedtools intersect and subtract commands.

## Immunostaining and quantitative image-based cytometry (QIBC) analysis

2000 HCT116 cells seeded in glass-bottom 384-well plate were pulse-labeled with 10 μM EdU for 30 min before harvest. For staining, cells were incubated in PBS-T buffer (0.2% Triton X-100 in 1 × PBS, PMSF, protease inhibitor cocktail, and phosphatase inhibitor cocktail) on ice for 5 min, followed by fixation with 2% paraformaldehyde. EdU staining was performed using the Click-iT EdU kit (ThermoFisher cat# C10634 or C10633) as per the manufacturer's protocol. Primary antibodies were diluted in PBS with 0.1% BSA and incubated at room temperature for 3 h. After washing with PBS-Tween20 (0.01%), plates were incubated with secondary fluorescently labeled antibodies (Alexa fluorophores, ThermoFisher cat# A11008, A21235) for 1 h. DAPI (0.5 μg/ml) staining was performed for 5 min at room temperature.

Image Acquisition and Analysis: After adding 50 μl of PBS to wells and sealing the plate, images were captured using Yokogawa CV7000 high-throughput confocal spinning disk microscope using a 60X water immersion lens (NA 1.2) using a scanning microscope. A minimum of 4 replicates were analyzed, with 20–30 images per condition capturing 1000–3000 cells each. Images were analyzed using the Columbus 2.7–2.9 high-content imaging analysis software (Revvity). First, nuclei were segmented using the DAPI staining image. Partial nuclei located at the edge of the image were excluded from subsequent steps of the analysis. This was followed by quantification of pixel intensities for EdU, RPA2 and H2AX. Well-level results from the Columbus image analysis were analyzed using R and data were plotted using ggplot2, facilitating visualization and quantification of replication stress and DNA damage signaling dynamics across cell populations.

## Proximity ligation assay (PLA)

Proximity ligation assays were carried out using the Duolink kit (Sigma Aldrich, Cat# DUO92101) following the manufacturer's instructions. The cells were seeded and grown on coverslips in a 6-well plate. After an overnight incubation, cells were pre-incubated with EdU before being fixed. Fixation was done with 4% paraformaldehyde in PBS for 15 min at 4 °C, followed by permeabilization with 0.25% Triton X-100 in PBS for 15 min at 4 °C. The coverslips were then blocked using the Duolink blocking solution. Primary antibodies were applied at a 1:500 dilution in Duolink antibody diluent and incubated at room temperature for 1–2 h. The PLA minus and plus probes were diluted 1:5 in the provided buffer, and 50 μL of the probe reaction was added to each coverslip and incubated for 1 h at 37 °C. The coverslips were then washed twice with buffer A. Next, the ligation buffer was diluted 1:5 in water, and ligase was added at a 1:30 dilution. The reaction was incubated at 37 °C for 30 min, followed by two washes with buffer A. The amplification buffer was also diluted 1:5 in water, and the polymerase was added at a 1:80 ratio. The amplification reaction was incubated at 37 °C for 100 min and then quenched by two washes with buffer B. Coverslips were mounted on slides using a DAPI-containing mounting medium. Yokogawa CV7000 high-throughput confocal spinning disk microscope using a 60X water immersion lens (NA 1.2) using a scanning microscope and analyzed with ImageJ (Fiji).

## Growth curve

Cells were seeded into 384-well white plate (Revvity Health Science, cat# 6007680) in octuplicate (8 wells per cell line) at a concentration of 50 cells per well in 60 μl medium. Medium was added to empty wells to avoid samples evaporation. Plates were then centrifuged at 1200 RPM for 2 min and cells were allowed to adhere overnight. At the indicated times, cells were incubated 20 minutes at room temperature and then 20 μl of CellTiter-Glo 2.0 reagent (Promega #G7570) was added. Plates were kept in the dark at room temperature for 20 additional minutes (10 min on a shaker and 10 min without shaking) and the luminescence was recorded using a SpectraMax i3x reader and the SoftMax Pro 7 software (Molecular Devices). Averages of octuplicates were calculated and ratios using t = 0 (1 day after plating) were determined. Graph generated using Prism 10.1.1 (GraphPad software).

## Reporting summary

Further information on research design is available in the Nature Portfolio Reporting Summary linked to this article.

## Data availability

The ChIP- and NS-sequencing raw and processed data generated in this study have been deposited in NCBI's Gene Expression Omnibus with the GEO Series accession number GSE276856, GSE247469 and GSE172417. Source data includes raw data for plots, an example gating strategy for flow cytometry and uncropped blots. Source data are provided with this paper.

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

## Acknowledgements
We thank the CCR core sequencing facility headed by Bao Tran and Jyoti Shetty for expert help with nascent strand DNA sequencing from fibroblast and cancer cell lines. We thank Michael Kruhlak, Langston Lim, and Andy Tran (Confocal Microscopy Core Facility, CCR, NCI, NIH) and Gianluca Pegoraro (HiTF Core Facility, CCR, NCI, NIH) for expert technical assistance in microscopy and quantification. We thank the CCR Genomics Core led by Liz Conner for the expert help with ChIP-sequencing, especially Madeline Wong and Steven Shema for their quick and excellent work. We thank Drs. Anjali Dhall, Anagh Ray, Gheorghe Chistol, Scott Berger and Ms. Elyse Hwang for critical reading of the manuscript and helpful comments. This work utilized the computational resources of the NIH HPC Biowulf cluster (http://hpc.nih.gov).

## Author contributions
Conceptualization: B.L.T., M.I.A., Methodology: B.L.T., C.E.R., H.F., R.S., N.A.K., L.S.P., M.I.A., Investigation: B.L.T., C.E.R., H.F., N.A.K., S.Z.Z., L.S.P., V.A.B., M.I.A., Visualization: B.L.T., N.A.K., L.S.P., S.Z.Z., M.I.A., Funding acquisition: M.I.A., Project administration: M.I.A., Supervision: M.I.A., Writing – original draft: B.L.T., M.I.A. Writing – review & editing: C.E.R., H.F., R.S., N.A.K., V.A.B., M.I.A.

## Funding
National Cancer Institute ZIA BC010411 (MIA). Open access funding provided by the National Institutes of Health.

## Competing interests
The authors declare no competing interests.
