## [Transparent Peer Review file · Nature Communications]

Selective interactions at pre-replication complexes categorize baseline and dormant origins

Corresponding Author: Dr Mirit Aladjem

Version 0:

Reviewer comments:

Reviewer #1

(Remarks to the Author)

This manuscript documents the impact of pre-RC binding by MTBP and RecQL4 on baseline and dormant origin firing. The paper is important. The paper contains new mechanistic insights. However, the paper is very difficult to understand as it is very dense and lacks some information that would be helpful for the reader not thoroughly acquainted with the authors' technology.

The following questions would help clarify the manuscript for this reader:

There were two sets of figure legends in the manuscript. I used the ones marked "Corrected Figure Legends."

Could the authors explain how the baseline and dormant origins used in this manuscript were defined. I assume it was in a previous manuscript? There is no mention of where these sequences are or how they were scientifically described. The reader has to understand this as its a variable for endpoint in every figure.

The text needs to be more descriptive.

Could the authors clarify that the novelty in the paper is the binding of RecQL4 to dormant origins. I think the authors need to be more precise in defining what has been published previously and what they show here, for the first time.

The authors should discuss the seemingly contradictory role of ATR on dormant origin firing in the literature.

Could the authors speculate as to what RecQL4 is doing in the way of activity?

It's all rather confusing.

Reviewer #2

(Remarks to the Author)

Review of NCOMMS-24-60205 (Aladjem)

In this work, the authors propose that phosphorylation of RecQL4 differentially affects the binding of Treslin-MTBP to baseline versus dormant origins. In particular, phospho-RecQL4 would prevent the binding of Treslin-MTBP to dormant origins. Upon replication stress, phospho-RecQL4 would somehow promote the dissociation of Treslin-MTBP from chromatin and later RecQL4 (presumably dephosphorylated) would allow the binding of Treslin-MTBP to both baseline and dormant origins. The authors are addressing an important topic and have formulated an interesting model. However, the supporting data has numerous problems. In addition, the overall model does not seem very plausible.

Specific Points.

Figure 1A. The experimental setup is not ideal. The “wild-type” cells would actually contain a mixture of wild-type and mutant Mcm2-S139A. I am concerned about potential dominant-negative effects.

Figures 1B and C. Why don't the authors show the blot and peaks for Treslin?

Figure 1B. Others have shown that Treslin-MTBP binds to chromatin in G1 and gradually leaves during S-phase. Phosphorylation of Mcm2 on S139 would presumably occur at S-phase. It is implausible that no Treslin-MTBP binds to chromatin in the presence of the Mcm2-S139A mutant.

It is difficult to understand how phosphorylation of RecQL4 would differentially affect its binding to baseline versus dormant origins. It would be helpful for the authors to summarize what is known about the role of this phosphorylation. For example, it has been proposed that it has a role in DNA repair at S/G2. There also needs to be a more thorough characterization of this phosphorylation in the context of the present work. For example, how does the expression of wild-type or mutant SIRT1 affect this phosphorylation?

Replication stress would affect CDK activity, which could potentially affect the phosphorylation of RecQL4 on S89 and possibly indirectly alter the phosphorylation of Mcm2 on S139. These issues should be addressed.

Phosphorylation of Mcm2 on S139 seems less often studied than some of the other DDK sites. Why was this site chosen?

Figure S2A. As mentioned elsewhere, several groups have shown that Treslin-MTBP binds to chromatin in G1. The discrepancies need to be cited and explained.

Recent studies have suggested that RecQL4 acts at a relatively late stage after DONSON, which does not seem to fit the RecQL4 acting upstream of Treslin-MTBP, as the authors are proposing. This issue needs to be discussed.

The authors should explain how they classify baseline versus dormant origins. My impression is that the distinction is not as clearcut as implied by this work. Are all initiation sites included in these analyses?

As discussed for the S139 phosphorylation, it is surprising that Treslin-MTBP and RecQL4 could not bind to chromatin in cells harboring the Mcm2-S108A mutant. It would be helpful if the authors provided supporting immunoblots of chromatin at different stages of the cell cycle.

Figure 3E. I could not find the protocol for the immunoprecipitation experiment. Mcm proteins are notoriously difficult to immunoprecipitate properly as they encircle DNA. The experiment could also use a loading control or some reference protein. Have the authors blotted for Mcm2 protein? There is no indication that the experiment was done more than once, and there is no quantitation or documentation of statistical significance (as is the case for a good deal of the data in the paper).

If phospho-RecQL4 inhibits the binding of Treslin-MTBP to dormant origins, the implication is that RecQL4 would have to undergo dephosphorylation upon replication stress so that it could promote initiation after dissociation and rebinding. Is there any evidence of this occurring?

I found the Discussion somewhat confusing. The authors could articulate their model more precisely.

Reviewer #3

(Remarks to the Author)

Thakur et al. present intriguing work on the control of origin firing in human cells. The authors built on their previous work on the involvement of SIRT1 and Mcm2 phosphorylation in the distinction between what they call baseline and dormant origins. This is important because it has been firmly established that the firing of dormant origins in replication stress conditions can rescue complete genome replication in cells experiencing replication stress and can prevent genetic alterations, whilst also being a source of replisome collapse. However, the cellular and molecular mechanisms that distinguish and control dormant origin firing have remained largely unknown. Thus, understanding dormant origins is important for both understanding cancer development and developing therapy.

Here, the authors characterise how the origin firing factors MTBP-Treslin and the phosphorylation of RecQL4 at serine 89 are involved. They establish that MTBP-Treslin dynamically associate and dissociate with origins dependently on whether the origins become active or not. This suggests that MTBP-Treslin do not sit on pre-RCs waiting for activation, but their arrival coincides with and may trigger origin firing. Phospho-RecQ4-S89 levels are high on silent dormant origins and decreases when these origins become activated. Phospho-RecQ4-S89 appears to be specific for dormant origins as it is not enriched on silent baseline origins (e.g. late origins in early S). The link of these regulations with SIRT1 and phospho-Mcm2-S139 is investigated.

The study presents technically sophisticated experiments that show largely clear and strong effects. Given that additional control experiments and analyses will be presented in a revised manuscript the presented data support the conclusions drawn and the work will be suitable for publication in Nature Communications.

Issues to be addressed:

- All cell lines should be described to some extent with new lines described in more detail.

Mention briefly how previously published lines were made. How was the transgene integrated and is it expressed using a constitutive strong promoter? Add references for more detail.

- It is important that chromatin and origin binding experiments are controlled for whether signals are pre-RC-dependent. The authors mention several times that they think of MTBP-Treslin and pRecQ4-S89 being pre-RC-associated. This control is important especially because MTBP and RecQ4 have been shown to bind chromatin off origins. Moreover, the fact that RecQ4-pS89 occurs at early origin in mid/late S phase suggests that not all binding detected is pre-RC-mediated. This control could be done using licensing-deficient cells generated by treating Mcm2-AID cells with IAA or by using siRNAs against Cdc6 or Cdt1. Alternatively, extensive PLA analyses could be done showing proximity with Mcm2-7. I prefer using licensing-deficient cells.

- Release from aphidicolin replication arrest is used to show how replication (Fig. 3C/D), exclusiveness of MTBP and RecQ4-pS89 (Fig. 3D), origin analysis (Fig. 4) and DNA damage/replication stress/survival (Fig. 5) are affected by aphidicolin (replication stress), and how this dynamically changes during recovery from replication stress. I agree that showing the dynamic behaviour adds value compared to analysis of just one time point. However, the two most important samples are missing, which are cells in the aphidicolin arrest (0h) and vehicle (presumably DMSO) treated cells. The authors please add these samples to the analyses. Cells 1h after release cannot replace the aphidicolin (0h) sample.

- Fig 1B:

A rescue experiment with Mcm2-WT is essential, because, as I understand, the mutation was not introduced genomically but using a transgene.

Cell cycle data with the experiment must be given to exclude that the differences are cell cycle defect

Describe how the chromatin fraction was isolated.

Show if signals depend on pre-RCs.

- 1C: Label lower experiment (with green peaks). Which conditions were used?

- 1D:

Essential is to clarify if ChIP signals shown here are dependent on pre-RCs.

Describe briefly how Initiation was determined. I assume using EdU-seq reads.

- 1E:

Describe and label all elements shown in the figure.

- S1A:

Add a rescue experiment with Mcm2-WT

Give numbers of all cell cycle populations, not only S phase.

Make a statement about expression levels of mutants vs WT and vs endogenous protein. Do so for all other cell lines comparing mutants and WT.

- S1B:

Give numbers of all cell cycle populations, not only S phase.

Make a statement about expression levels of transgenes.

You call cells 'isogenic'. I am not sure this expression is appropriate as it involves that transgenes were integrated in the same locus.

- S1C:

PPase shows dependency on phosphorylation. Use the A mutant to show that ab detects specifically the S89 site.

- S2A:

Test pre-RC-dependency of signals.

- 2E:

Discuss in more detail the association of pRecQ4-S89 with early dormant origins in mid and late time points. This association must be independent of pre-RCs, because there are no pre-RCs anymore on early origins.

- 2F:

This figure should involve a potential binding of the phospho-RecQ4 to origins DNA in the absence of pre-RCs as it occurs at least on early dormant oris at late time points. Are there 2 modes of origin binding, pre-RC-dependent and independent?

- Fig 3:

Mention how the 108A cell line was made (albeit for a previous project). Also describe a bit better some basic characteristics: how well does it replicate and how efficient which origins fire in normal and replication stress conditions.

The authors please consider to show the origin analysis by ChIP (Fig 4) before the complex formation experiment in Fig 3E. I find this part very hard to follow.

- Fig 3B:

Do dormant origins fire despite the absence of RecQ4-pS89 in Mcm2-S108A cells (no Ex527)? Mention and discuss. If they do not fire this means that absence of phospho-RecQ4 is not sufficient for dormant origin firing.

The ATR-Chk1 axis is known to restrict dormant origin firing. However, the authors suggest that ATR is required for Mcm2-S108-mediated dormant origin firing. Is Mcm2-S108 phosphorylated ATR-dependently in this set-up? This issue needs to be mentioned and discussed. ATR inhibitor could be used to test S108 phosphorylation by ATR. ATRi in combination with Mcm2-S108A could be used to test if ATRi-mediated dormant ori firing is also Mcm2-S108-dependent.

3E:

The experiment is insufficiently described:

- How was the chromatin fraction made?

- How were pMCM2-associated proteins bound to the ab-coupled beads solubilised for subsequent IPs?

- What were the levels of proteins in the supernatant after PCNA-depletion of replisomes? Show the IP-PCNA-flow through as input for subsequent IPs.

The authors intend to show that MTBP and RecQ4-pS89 occupy separate Mcm2-7 fractions. But they use a phospho-Mcm2-S139 antibody for IP, the eluate of which is used for subsequent IPs. Because Mcm2-pS139 and RecQ4-pS89 were shown to be largely exclusive (Fig 1), anti-Mcm-IP (not phospho-MCM-IP) should have been used to eliminate this bias. The authors please describe this more clearly, including what can be concluded from the experiment in light of my point raised.

Fig. 4B/D:

The ChIP data should be displayed the same way it has been shown in previous figures: separated for baseline and dormant origins like in Fig 1D. I can see that the plots shown here constitute a richer data source. However, this comes at the expense of clarity. If the authors see a benefit, the dot plots can be additionally shown, either in the main or supplementary figure. If the authors choose to show dot plots they please also show other ChIP data additionally as dot plots to keep data display consistent.

Here, only baseline origins are shown. Dormant origins should also be shown in the main figure.

The data shown should be more specifically described in the text. The dot plots are quite complicated to read.

Fig. 4C:

The RecQL4 cell lines need to be described characterised in more detail. How were the transgenes integrated? Were clones selected? were expression levels of transgenes similar? Did different clones behave similarly? How well did the cells grow and replicate, particularly the RecQL4 knock out cells. I am surprised they are alive. Are they strongly growth-compromised?

Replication from dormant origins in RecQ4-89A cells as well as Mcm2-S139 phosphorylation status should be shown to complete the dataset.

Fig 5B:

Labelling unclear: release from a 1h treatment with 10uM APH

Fig. S5D:

The connection between telomere length and replication stress must be described in a bit more detail.

- pS89-RecQ4 was implicated in positive firing regulation (Shin et al. JBC 2019). Please discuss.

Fig 5E:

pre-RC-dependency of factor binding to origins was not tested.

Discuss why pRecQL4-deficient cells have strong difficulties to recover from replication stress. After all they do fire dormant origins so these should rescue forks stalled by aphidicolin. Is it because excess dormant origin firing increases fork collapse? So this might be a special situation when forks are strongly stalled with high aphidicolin. If forks were slow in low aphidicolin the effect of mutant RecQ4 might not be as strong. Is this worth discussing?

Version 1:

Reviewer comments:

Reviewer #1

(Remarks to the Author)

The authors have made a huge effort to address all reviewers' comments. I think the manuscript offers a significant advance. It is complex, but this reflects the methodology.

I find that the authors have addressed both my and Reviewer number 2's critique in depth. The authors show in several experiments that a single point mutation in the MCM complex blocks the binding of the Treslin-MTBP complex to chromatin. I can't see a flaw in their experimental design. That reviewer number 2 is surprised is irrelevant, it's what the reviewers show, and it appears definitive to me.

Their model is that pRecQL4 distinguishes between dormant and baseline origins and their data supports this model. Specifically, pRecQL4-S89 clearly binds dormant origins in the presence of SIRT1. This is repeated in Figure 2E. Clearly pRecQL4-S89 is binding a subset of origins. In Figure 3B, this binding is shown to be dependent on MCM2 S108, which is a known phosphorylation site. I do not see any ambiguity in these datasets. I could still question what the baseline and dormant origin sequences are, but pRecQL4-S89 binding defines the "dormant" origins precisely. Same for Figure 4C, 5A, 5B.

Figure 4E: can the authors write 2nd, not 2st.

There is an enormous amount of data in the paper. The authors have done several additional experiments after review. There is no suggestion that the data is contrived or uncontrolled. It's possible their interpretation is wrong or confounded by something we don't understand, but I don't think there is any justification to reject the paper.

Reviewer #2

(Remarks to the Author)

The authors have not addressed my concerns. Others have shown that Treslin-MTBP binds in G1 to chromatin and gradually leaves during S-phase. It is also very surprising that single point mutations in the MCM complex would block the binding of the Treslin-MTBP complex (about 800 kD) to chromatin. There is not a good explanation for how pRecQL4 would distinguish between dormant and baseline origins. The overall model seems implausible. The Debatisse group has proposed another model for the control of dormant origins involving regulation of TopBP1 that in principle seems more logical (Koundrioukoff et al., bioRxiv, <https://doi.org/10.1101/2023.11.29.569233>). I would recommend rejection of this manuscript.

Reviewer #3

(Remarks to the Author)

The revisions are excellent and I fully support publication of the manuscript.

minor points:

line 89: Add 'Although RecQL4 is essential for replication origin firing in *Xenopus* egg extracts and chicken DT40 cells...'

Fig2F/line 284: '...it is likely that pRecQ4 binding occurred after those dormant origins were passively replicated...'

The model in 2F seems inconsistent to me with this sentence as 2F shows RecQ4 bound to pre-RCs. Please clarify.

Fig S1D: A colour code seems to be missing in the Diagram. I do not see which circle belongs to which condition.

Fig S3B: Align labelling in right panel with the lanes.

Point-by-Point Response to Reviewers' Comments:

Reviewer #1 (Remarks to the Author):

Comment: This manuscript documents the impact of pre-RC binding by MTBP and RecQL4 on baseline and dormant origin firing. The paper is important. The paper contains new mechanistic insights. However, the paper is very difficult to understand as it is very dense and lacks some information that would be helpful for the reader not thoroughly acquainted with the authors' technology.

Response: We thank the reviewer very much for this thoughtful evaluation. We appreciate the reviewer's evaluation that the paper is important, and we agree that it is important to clarify the methodological aspects. In the revision, we have included detailed descriptions of some of the uncommon experimental approaches used in the study and have provided more background about specialized aspects of the study. We believe that the revised version will be clearer and more readable and that it has improved the paper

Comment: The following questions would help clarify the manuscript for this reader: There were two sets of figure legends in the manuscript. I used the ones marked "Corrected Figure Legends."

Response: We apologize for the confusion, the revision contains a single copy of the figure legends.

Comment: Could the authors explain how the baseline and dormant origins used in this manuscript were defined. I assume it was in a previous manuscript? There is no mention of where these sequences are or how they were scientifically described. The reader has to understand this as its a variable for endpoint in every figure.

Response: We thank the reviewer for pointing out this issue. The reviewer is correct, we categorized baseline and dormant origins based on our previously published observations, specifically, the differential activation of replication origins in SIRT1-Wild Type (WT) and SIRT1-Mutant cells. We made the following changes in the revision:

1. In the Introduction section (starting at the last paragraph of page 3), we provide a summary of these observations (reported in two previously published papers, Utani et al., PMID: 28549174 and Thakur et al., PMID: 35524559, cited as references 41 and 15).

	Origin count (paired comparison)		
	Total	Common (baseline)	Newly active (dormant)
WT-SIRT1	121054	121054	-
WT-SIRT1+APH	130134	104571	29108
Mut-SIRT1	151194	111796	47209

2. In the Results section, we clarify that origins detected in cells harboring WT-SIRT1 are referred to as baseline origins, and origins that are activated in addition to the baseline origins in cells harboring the SIRT1 mutant are defined as dormant origins (first paragraph, page 5).

3. In Supplementary Figure 1D (also shown on the left), we now show that dormant origins identified by their ability to initiate replication in cells harboring mutant, but not WT, SIRT1 demonstrate a high concordance with dormant

origins identified based on their inactivation during unperturbed growth and activation by replication stress.

Comment: The text needs to be more descriptive.

Could the authors clarify that the novelty in the paper is the binding of RecQL4 to dormant origins. I think the authors need to be more precise in defining what has been published previously and what they show here, for the first time.

Response: We thank the reviewer for this suggestion. We have revised the Abstract to emphasize the role of RecQL4 in origin dormancy and in the cellular recovery from replication stress. In addition, we have expanded the Introduction (page 3, 2nd paragraph) to clarify that although RecQL4 was known as a pre-RC binding protein, its role in origin initiation is yet unclear.

Comment: The authors should discuss the seemingly contradictory role of ATR on dormant origin firing in the literature.

Response: We thank the reviewer for this important suggestion. In the revision, we provide more details about the role of the ATR pathway and its interactions with the SIRT1 pathway in the activation or suppression of replication initiation (Introduction, starting at the last paragraph of page 3). We also discuss how RecQL4-mediated suppression of origin activity relates to ATR-mediated activation of dormant origins via MCM2-S108 phosphorylation (Discussion, starting at the last paragraph on page 17).

Comment: Could the authors speculate as to what RecQL4 is doing in the way of activity? It's all rather confusing.

Response: We thank the reviewer for this suggestion, it is a very interesting question. We have expanded the Discussion section to provide more details and hypotheses regarding the role of RecQL4 activity in the regulation of DNA replication and repair (starting on page 18).

Reviewer #2 (Remarks to the Author):

Review of NCOMMS-24-60205 (Aladjem)

Comment: In this work, the authors propose that phosphorylation of RecQL4 differentially affects the binding of Treslin-MTBP to baseline versus dormant origins. In particular, phospho-RecQL4 would prevent the binding of Treslin-MTBP to dormant origins. Upon replication stress, phospho-RecQL4 would somehow promote the dissociation of Treslin-MTBP from chromatin and later RecQL4 (presumably dephosphorylated) would allow the binding of Treslin-MTBP to both baseline and dormant origins. The authors are addressing an important topic and have formulated an interesting model. However, the supporting data has numerous problems. In addition, the overall model does not seem very plausible.

Response: We thank the reviewer for the evaluation of our submission and for the thoughtful comments. In the revision, we provide additional data, expand the model and address all the reviewers' concerns, below.

Specific Points.

Comment: Figure 1A. The experimental setup is not ideal. The “wild-type” cells would actually contain a mixture of wild-type and mutant Mcm2-S139A. I am concerned about potential dominant-negative effects.

Response: We thank the reviewer for pointing out this issue. To address this concern, the revision includes a new control experiment in which we utilized the parental MCM-mAID, and its counterpart clones complemented with WT-MCM2 as well as MCM2-S139A (also suggested by Reviewer #3). Complementation of MCM-mAID with the WT-MCM2 restored Treslin, MTBP and pRecQL4 chromatin association whereas complementation with MCM2-S139A did not. The result of this additional experiment is shown on the left and in figure 1D of the revision.

Figure 1D: The abundance of pMCM2-S139, MCM2, Treslin, MTBP and pRecQL4-S89 in whole cell and chromatin extracts from HCT116 cells harboring MCM2-mAID along with MCM2-WT and MCM2-S139A with and without auxin (IAA, 16hr).

Comment: Figures 1B and C. Why don't the authors show the blot and peaks for Treslin?

Response: Thank you, sorry for the omission. In the revision, the blots and peaks for Treslin, as well as MTBP, are shown below and in figure 1, D (above) and E (left).

Figure 1E: Chromatin binding sites of phospho-MCM2 S139 (pMCM2-S139), MTBP, RecQL4, and phospho-RecQL4 S89 (pRecQL4-S89) were mapped by chromatin immunoprecipitation followed by sequencing (ChIP-Seq) in WT^{SIRT1} and Mut^{SIRT1} cells. Shown are IGV screenshots of the chromosome 1 genomic region containing baseline and dormant origins, displaying averaged ChIP-seq coverages from two biological replicates.

Comment: Figure 1B. Others have shown that Treslin-MTBP binds to chromatin in G1 and gradually leaves during S-phase. Phosphorylation of Mcm2 on S139 would presumably occur at S-phase. It is implausible that no Treslin-MTBP binds to chromatin in the presence of the Mcm2-S139A mutant.

Response: We thank the reviewer for this comment, which prompted us to explore Treslin and MTBP binding during the G1 and S phases in more detail. In the revision (shown below and in supplementary figure 2, A-D), a new Figure, we present an experiment in which we followed the association of MTBP, Treslin and pRecQL4 with chromatin throughout the G1 and S phases in cells released from a nocodazole block. Our results suggest, in line with previous findings, that MTBP, Treslin and pRecQL4 are recruited to chromatin at a low level in a diffuse pattern during the G1 phase. At the onset of S phase, these proteins exhibit increased chromatin binding and converge to localize primarily at replication origins. Our

Supplementary Figure 2: A: Cells harboring WT-MCM2 and S139A-MCM2 were exposed to nocodazole, and released for the indicated time period. Cell cycle distributions were by flow cytometry using an EdU incorporation/DAPI assay. Top represents DNA content (grey). B: An immunoblot measuring the abundance of pre-RC components in the chromatin bound fraction (Chr) of HCT116 cells at the indicated timepoints during (0hr) and after release from a G2/M block induced by Nocodazole (Noc rel) in cells expressing MCM2-WT and MCM2-S139A post IAA (16hr along with nocodazole) treatment. C: Heatmaps depicting the distribution of MTBP and Treslin binding sites centered on baseline and dormant origins at the indicated timepoint after release from a nocodazole-induced G2/M block as described in the legend to panel A. Baseline and dormant origins were stratified as described in Figure 1, A and B. D: Quantification of the extent of origin binding of MTBP and Treslin to baseline and dormant origins and to non-origin regions following a nocodazole-induced G2/M block (data from panel B).

Figure 1D: The abundance of pMCM2-S139, MCM2, Treslin, MTBP and pRecQL4-S89 in whole cell and chromatin extracts from HCT116 cells harboring MCM2-mAID along with MCM2-WT and MCM2-S139A with and without auxin (IAA, 16hr).

immunoblot analysis (left and in figure 1D in the revision) detects the robust chromatin binding, which occurs in S-phase and does not occur in the MCM2-S139A mutant, whereas the low-level, diffuse binding at the G1 phase is below the detection threshold.

This finding is in line with the observed high association with origins at the onset of S-phase, and the decreased chromatin association of MTBP and Treslin as S-phase progresses, which was also reported by others. Because MTBP and Treslin bind replication origins prior to initiation, their chromatin association with replication origins appears at the highest levels during the early stages of S-phase

Figure 2D: ChIP-seq using MTBP antibodies was performed in cells collected at the indicated times after double thymidine block. The heatmaps show MTBP and pRecQL4-S89 signal strengths in early, mid, and late origins.

Supplementary Figure 2G: ChIP-seq was performed in cells collected at the indicated times after double thymidine block to map interactions of baseline origins with Treslin. The heatmaps illustrate signal strength on early, mid, and late origins.

(when they associate with early-replicating origins) and subsequently decreases (when they bind late-replicating origins, which are less numerous). The kinetics of origin binding for MTBP and Treslin during S-phase are shown above and in the revised Figure 2D (MTBP) and supplementary figure 2G (Treslin).

Comment: It is difficult to understand how phosphorylation of RecQL4 would differentially affect its binding to baseline versus dormant origins. It would be helpful for the authors to summarize what is known about the role of this phosphorylation. For example, it has been proposed that it has a role in DNA repair at S/G2. There also needs to be a more thorough characterization of this phosphorylation in the context of the present work. For example, how does the expression of wild-type or mutant SIRT1 affect this phosphorylation?

Response: We thank the reviewer for this suggestion. In the revision, we summarize what is known about RecQL4 phosphorylation as suggested (page 3, middle paragraph). We also include data suggesting that the levels of RECQL4 phosphorylation are similar in cells with and without active SIRT1 (shown on the left and in supplementary figure 1G of the revision). RecQL4 deacetylation by SIRT1 has been reported, and this observation is put in context in the revised discussion (page 19, top paragraph).

Supplementary Figure 1G: Abundance of phosphorylated SIRT1-T530, total SIRT1, pRecQL4-S89 and RecQL4 in whole cell extracts of the cells harboring WT and mutant (H363Y) SIRT1.

Comment: Replication stress would affect CDK activity, which could potentially affect the phosphorylation of RecQL4 on S89 and possibly indirectly alter the phosphorylation of Mcm2 on S139. These issues should be addressed.

Response: Thank you for mentioning this important fact. In the revision, we discuss the fact that both RecQL4 and Treslin are CDK substrates (pages 18-19) and the potential impact of altered kinase activity on helicase activation. We also emphasize that we cannot rule out indirect effects of MCM2 phosphorylation on CDK-mediated phosphorylation of RecQL4.

Comment: Phosphorylation of Mcm2 on S139 seems less often studied than some of the other DDK sites. Why was this site chosen?

Response: We thank the reviewer for raising this point. In the revision, we expanded the Introduction (page 2-3) to summarize the primary finding of our previous publication (Thakur et al., NAR 2022, PMID: 35524559, reference 15), in which we reported that S139 on MCM2 is primarily phosphorylated by DDK and that this phosphorylation, which is essential for the initiation of DNA replication, is markedly stronger at baseline origins than

at dormant origins. These observations provide a rationale for focusing on that phosphorylation site in the current submission. In the revised Results section we provide a detailed explanation of the choice of this modification (page 6, top paragraph).

Comment: Figure S2A. As mentioned elsewhere, several groups have shown that Treslin-MTBP binds to chromatin in G1. The discrepancies need to be cited and explained.

Response: We thank the reviewer for raising this point. We observed a diffused pre-RC proximal binding for both MTBP and Treslin in G1, which were enhanced at pre-RCs (replication origins sites) as cells progressed toward the S-phase (shown below and in supplementary figure 2, A-D of the revision). We also noted that MTBP-Treslin binding is lower at genomic regions not associated with origins.

Supplementary Figure 2: A: Cells harboring WT-MCM2 and S139A-MCM2 were exposed to nocodazole, and released for the indicated time period. Cell cycle distributions were by flow cytometry using an EdU incorporation/DAPI assay. Top represents DNA content (grey). B: An immunoblot measuring the abundance of pre-RC components in the chromatin bound fraction (Chr) of HCT116 cells at the indicated timepoints during (0hr) and after release from a G2/M block induced by Nocodazole (Noc rel) in cells expressing MCM2-WT and MCM2-S139A post IAA (16hr along with nocodazole) treatment. C: Heatmaps depicting the distribution of MTBP and Treslin binding sites centered on baseline and dormant origins at the indicated timepoint after release from a nocodazole-induced G2/M block as described in the legend to panel A. Baseline and dormant origins were stratified as described in Figure 1, A and B. D: Quantification of the extent of origin binding of MTBP and Treslin to baseline and dormant origins and to non-origin regions following a nocodazole-induced G2/M block (data from panel B).

Comment: Recent studies have suggested that RecQL4 acts at a relatively late stage after DONSON, which does not seem to fit the RecQL4 acting upstream of Treslin-MTBP, as the authors are proposing. This issue needs to be discussed.

Response: Thank you, we agree, in the revision, we have expanded the Introduction (page 3, 2nd paragraph) and discussion (page 19, top paragraph) to summarize pertinent literature characterizing the association of RecQL4 with pre-RC and functional studies assigning

Figure 2E: ChIP-seq using pRecQL4-S89 antibody was performed in cells collected at the same time points as in panel C. The heatmaps show pRecQL4-S89 signal strengths in early, mid, and late chromatin centered on origins stratified as in figure 2 panel C. pRecQL4-S89, the left panel shows binding to baseline origins and the right panel shows binding to dormant origins

both positive and negative roles to RecQL4.

Specifically, the proposed role of RecQL4 in separating DONSON-containing pre-RC subunits to initiate replication was not yet explored in the context of RecQL4 phosphorylation (we have ongoing discussions about this possibility with colleagues studying the RecQL4-DONSON interactions, and we hope to collaboratively explore the role of phosphorylation directly in future experiments). As we mention in the Discussion (page 19), it is possible that RecQL4 phosphorylation hampers this newly reported function of RecQL4.

Importantly, we observed that the phosphorylated form of RecQL4 is bound to dormant origins in late S after replication (shown on the left, and in figure 2E of the revision), suggesting that pRecQL4 might have a separate, pre-IC independent role in regulating origin dormancy.

Comment: The authors should explain how they classify baseline versus dormant origins. My impression is that the distinction is not as clearcut as implied by this work.

Response: Thank you for this comment, which was also raised by Reviewer #1. In the revision, we clarify that baseline and dormant origins were categorized based on the differential activation of replication origins by SIRT1. Origins identified in cells harboring WT-SIRT1 are referred to as baseline origins, whereas dormant origins are defined as those origins that are activated in addition to the baseline origins in cells harboring the SIRT1 mutant. We also show that dormant origins called according to this classification demonstrate a high concordance with dormant origins identified based on their

	Origin count (paired comparison)		
	Total	Common (baseline)	Newly active (dormant)
WT-SIRT1	121054	121054	-
WT-SIRT1+APH	130134	104571	29108
Mut-SIRT1	151194	111796	47209

inactivation during unperturbed growth and activation by replication stress (shown on the left and in supplementary figure 1D of the revision).

Comment: Are all initiation sites included in these analyses?

Response: Yes, we have indeed included all the initiation sites in the analyses. We define baseline origins as all the origins active in WT-SIRT1 cells, while dormant origins are defined as origins activated in addition to the baseline origins cells harboring Mut-SIRT1 (Page 5, top paragraph).

Comment: As discussed for the S139 phosphorylation, it is surprising that Treslin-MTBP and RecQL4 could not bind

to chromatin in cells harboring the Mcm2-S108A mutant. It would be helpful if the authors provided supporting immunoblots of chromatin at different stages of the cell cycle.

Response: Thank you for this comment, we apologize for the misunderstanding. In the revision, we clarify that only the association with dormant origins was affected by the MCM2-S108 mutation. Specifically, these proteins were detected on chromatin, with MTBP and Treslin colocalized with baseline origins, and not with dormant origins, in cells harboring the MCM2-S108A mutant. Since immunoblots cannot distinguish between binding to dormant and baseline origins, we have provided this evidence using ChIP-Seq (left and in Supplementary Figure 3A).

Supplementary Figure 3A: HCT116 cells harboring MCM2WT and MCM2S108A were generated as illustrated in Figure 1C. Heatmaps are showing the binding of pMCM2-S139 and total RecQL4 to baseline and dormant origins with and without exposure to Ex527.

Comment: Figure 3E. I could not find the

protocol for the immunoprecipitation experiment. Mcm proteins are notoriously difficult to immunoprecipitate properly as they encircle DNA. The experiment could also use a loading control or some reference protein. Have the authors blotted for Mcm2 protein? There is no indication that the experiment was done more than once, and there is no quantitation or documentation of statistical significance (as is the case for a good deal of the data in the paper).

Response: We thank the reviewer for this suggestion and apologize for the omissions. We have added the sequential Re-IP protocol and indicated the number of repetitions in the Materials and Methods section (Supplementary Information). We digested DNA and RNA

using benzonase nuclease which allows to IP of MCM2. As suggested, the new experiment added in the revision also includes a loading control (shown below and in figure 4E of the revision). Finally, in the revised submission (page 6, 1st and 2nd paragraphs) we cite previous studies (from our group as well as others) showing that unlike immunoprecipitations of unmodified MCM proteins, immunoprecipitations with antibodies directed against MCM2 phosphorylated at specific residues, including pMCM2-S139 and pMCM2-S108, are efficient and reproducible.

Figure 4E: Replication stress facilitates an interaction between MTBP and pRecQL4. Left, experimental strategy. Right, Chromatin extracts from asynchronous HCT116 cells untreated or treated with APH (10 μ M for 1 hr) were used for sequential immunoprecipitation (IP) with PCNA to deplete progressing forks. The PCNA-depleted fraction was immunoprecipitated with pMCM2 antibodies to isolate pre-replication complexes. MTBP or pRecQL4 antibodies were used for further IPs, and the presence or absence of MTBP or pRecQL4 was detected by immunoblotting.

Comment: If phospho-RecQL4 inhibits the binding of Treslin-MTBP to dormant origins, the implication is that RecQL4 would have to undergo dephosphorylation upon replication stress so that it could promote initiation after dissociation and rebinding. Is there any evidence of this occurring?

Response: We thank the reviewer for raising this important point, which prompted us to measure the phosphorylation state of RecQL4-S89 during and after exposure to APH. We found that exposure to APH did not alter the total phosphorylation level of RecQL4, but the abundance of pRecQL4-S89 on chromatin was markedly increased during, and shortly after, exposure to APH. pRecQL4-S89 levels on chromatin subside during the subsequent recovery. These observations suggest that exposure to APH increased pRecQL4-S89 chromatin binding, followed by its dissociation from chromatin, in line with the observed

temporary redistribution of pRecQL4-S89 to baseline replication origins during these time points (left, and in Supplementary Figure 3B of the revision). The implications of this redistribution are discussed in the revision (page 17, top two paragraphs).

Supplementary Figure 3B: An immunoblot showing the abundance of the indicated proteins in total and chromatin extracts from HCT116 cells untreated, treated and released from exposure to APH (10 μ M for 1hr).

Comment: I found the Discussion somewhat confusing. The authors could articulate their model more precisely.

Response: Thank you for raising this point. In the revision, we have redrawn the model and revised the text to provide a more thorough discussion (shown below and in Figure 6).

Figure 6: A model illustrating dynamic interactions at replication origins during the recovery from replication stress. Left, in cells harboring WT-RecQL4, recruitment of the Treslin-MTBP complex marks a sub-group of baseline origins (e.g. origin 1) whereas pRecQL4-S89 (pRecQL4) associates with dormant origins (e.g. origin 2). When cells encounter replication stress, baseline origins can stall (red halo) but dormant origins, which do not initiate replication, are not affected and still maintain MCM complexes (green ring). Although pRecQL4 binding does not allow initiation from dormant origins under normal circumstances (see figure 2F), RecQL4 dissociates from origins when cells recover from replication stress and allows MTBP re-association with those origins. Upon the binding of MTBP and Treslin to dormant origins, these origins initiate replication to complete DNA synthesis. Right, in cells do not harbor RecQL4-S89 (either RecQL4 depleted or harboring the RecQL4-S89A substitution), MTBP and Treslin associate with both origins 1 and 2, and replication initiates from both origins when cells encounter acute replication stress. Under these conditions, replication from dormant origins cannot rescue the damage after replication stress is removed, preventing normal recovery and leading to the accumulation of ssDNA and subsequent DNA damage.

Reviewer #3 (Remarks to the Author):

Comment: Thakur et al. present intriguing work on the control of origin firing in human

cells. The authors built on their previous work on the involvement of SIRT1 and Mcm2 phosphorylation in the distinction between what they call baseline and dormant origins. This is important because it has been firmly established that the firing of dormant origins in replication stress conditions can rescue complete genome replication in cells experiencing replication stress and can prevent genetic alterations, whilst also being a source of replisome collapse. However, the cellular and molecular mechanisms that distinguish and control dormant origin firing have remained largely unknown. Thus, understanding dormant origins is important for both understanding cancer development and developing therapy.

Here, the authors characterise how the origin firing factors MTBP-Treslin and the phosphorylation of RecQL4 at serine 89 are involved. They establish that MTBP-Treslin dynamically associate and dissociate with origins dependently on whether the origins become active or not. This suggests that MTBP-Treslin do not sit on pre-RCs waiting for activation, but their arrival coincides with and may trigger origin firing. Phospho-RecQ4-S89 levels are high on silent dormant origins and decreases when these origins become activated. Phospho-RecQ4-S89 appears to be specific for dormant origins as it is not enriched on silent baseline origins (e.g. late origins in early S). The link of these regulations with SIRT1 and phospho-Mcm2-S139 is investigated.

The study presents technically sophisticated experiments that show largely clear and strong effects. Given that additional control experiments and analyses will be presented in a revised manuscript the presented data support the conclusions drawn and the work will be suitable for publication in Nature Communications.

Response: We thank the reviewer for the thorough reading and thoughtful evaluation of our study, and for the detailed, helpful suggestions. Below, we provide a detailed response outlining how we have addressed each of the reviewer's concerns.

Comment: Issues to be addressed:

- All cell lines should be described to some extent with new lines described in more detail. Mention briefly how previously published lines were made. How was the transgene integrated and is it expressed using a constitutive strong promoter? Add references for more detail.

Response: Thank you for suggestion. In the revision, we have expanded the Introduction and the Results sections (pages 2-5) to provide better descriptions of the cell lines used in this work, including citations and brief descriptions of previously published cell lines. We have also included descriptions of the technical details of cell line generations in the Materials and Methods section (Supplementary Information), including pertinent references.

Comment: - It is important that chromatin and origin binding experiments are controlled for whether signals are pre-RC-dependent. The authors mention several times that they think of MTBP-Treslin and pRecQ4-S89 being pre-RC-associated. This control is important

especially because MTBP and RecQ4 have been shown to bind chromatin off origins. Moreover, the fact that RecQ4-pS89 occurs at early origin in mid/late S phase suggests that not all binding detected is pre-RC-mediated.

This control could be done using licensing-deficient cells generated by treating Mcm2-AID cells with IAA or by using siRNAs against Cdc6 or Cdt1. Alternatively, extensive PLA analyses could be done showing proximity with Mcm2-7. I prefer using licensing-deficient cells.

Figure 1D, left: The abundance of pMCM2-S139, MCM2, Treslin, MTBP and pRecQL4-S89 in whole cell and chromatin extracts from HCT116 cells harboring MCM2-mAID along with MCM2-WT and MCM2-S139A with and without auxin (IAA, 16hr).

Response: We thank the reviewer for raising this important point. In the revision, we have tested our main conclusions for the dependence on pre-RC assembly by including appropriate controls. Like the reviewer, we prefer controls in which the cells were rendered licensing-deficient, primarily utilizing MCM2-mAID activation by Auxin to prevent the assembly of pre-RCs. Specific examples in which pre-RC dependence was tested are shown on the left and below (next page), corresponding to Figure 1D and Supplementary Figure 2, A-D of the revision.

Supplementary Figure 2, above: A: Cells harboring WT-MCM2 and S139A-MCM2 were exposed to nocodazole, and released for the indicated time period. Cell cycle distributions were by flow cytometry using an EdU incorporation/DAPI assay. Top represents DNA content (grey). B: An immunoblot measuring the abundance of pre-RC components in the chromatin bound fraction (Chr) of HCT116 cells at the indicated timepoints during (0hr) and after release from a G2/M block induced by Nocodazole (Noc rel) in cells expressing MCM2-WT and MCM2-S139A post IAA (16hr along with nocodazole) treatment. C: Heatmaps depicting the distribution of MTBP and Treslin binding sites centered on baseline and dormant origins at the indicated timepoint after release from a nocodazole-induced G2/M block as described in the legend to panel A. Baseline and dormant origins were stratified as described in Figure 1, A and B. D: Quantification of the extent of origin binding of MTBP and Treslin to baseline and dormant origins and to non-origin regions following a nocodazole-induced G2/M block (data from panel B).

Comment: - Release from aphidicolin replication arrest is used to show how replication (Fig. 3C/D), exclusiveness of MTBP and RecQ4-pS89 (Fig. 3D), origin analysis (Fig. 4) and DNA damage/replication stress/survival (Fig. 5) are affected by aphidicolin (replication stress), and how this dynamically changes during recovery from replication stress. I agree that showing the dynamic behaviour adds value compared to analysis of just one time point. However, the two most important samples are missing, which are cells in the aphidicolin arrest (0h) and vehicle (presumably DMSO) treated cells. The authors please add these samples to the analyses. Cells 1h after release cannot replace the aphidicolin (0h) sample.

Response: Thank you for pointing out this important control. In the revision, we included the 0 hr and vector controls, as suggested (shown below and in figure 4C-D of the revision).

Figure 4C: A heatmap showing binding of MTBP (left) and pRecQL4-S89 (right) to baseline and dormant origins in HCT116 cells collected immediately after exposure to APH (APH) or at indicated time points post-APH removal.

Figure 4D: Quantification of the fractions (percent) of baseline (top) and dormant (bottom) origin sequences binding to MTBP, pRecQL4, or both, at the indicated timepoint post-APH wash.

Comment: - Fig 1B:

A rescue experiment with Mcm2-WT is essential, because, as I understand, the mutation was not introduced genomically but using a transgene.

Response: Thank you, this control rescue with WT-MCM2 (also addressing a comment from reviewer #2) is included in the revision (shown on the left and in figure 1D).

Figure 1D: The abundance of pMCM2-S139, MCM2, Treslin, MTBP and pRecQL4-S89 in whole cell and chromatin extracts from HCT116 cells harboring MCM2-mAID along with MCM2-WT and MCM2-S139A with and without auxin (IAA, 16hr).

Comment: Cell cycle data with the experiment must be given to exclude that the differences are cell cycle defect

Supplementary figure 1E: Top, a cell cycle distribution of MCM2S139A cells with and without exposure to IAA. Bottom, quantification of at least 3 biological replicates.

Response: Thank you, this control was included in the revision (shown on the left and in supplementary figure 1E of the revision).

Comment: Describe how the chromatin fraction was isolated.

Response: Thank you, this description was added in Methods section the revision (Supplementary information, page 3).

Comment: Show if signals depend on pre-RCs.

Response: Thank you, the MCM2-mAID control was included in the revision (see above, figure 1D in the revision).

Comment: - 1C: Label lower experiment (with green peaks). Which conditions were used?

Response: Thank you and sorry for the omission. In the revision, we show NS-Seq and ChIP-Seq in separate panels (Figures 1B for NS-Seq and 1E for ChIP-Seq), both clearly labeled. The revised Materials and Methods section (Supplementary Information) described in detail the conditions utilized for ChIP-Seq.

Comment: - 1D:

Essential is to clarify if ChIP signals shown here are dependent on pre-RCs.

Response: Thank you, as mentioned above, we have tested all our reagents using mAID-MCM2, as suggested by the reviewer.

Comment: Describe briefly how Initiation was determined. I assume using EdU-seq reads.

Response: Thank you, we have included a brief description of the methodology (NS-Seq) in the revision (page 5 of the Method section).

Comment: - 1E:

Describe and label all elements shown in the figure.

Response: Thank you, we have labeled all the elements and also added corresponding Mut-SIRT1 data (revised figure 1G, below).

Figure 1G: Quantification of the chromatin binding patterns of MTBP, Treslin, pRecQL4-S89, pMCM2-S139 and RecQL4 shown in panel F (above). The bar plot shows the enrichment of ChIP-Seq signals at baseline and dormant origins in two biological replicates from cells harboring WT^{SIRT1} and Mut^{SIRT1} vs. input controls.

Comment: - S1A:

Add a rescue experiment with Mcm2-WT

Give numbers of all cell cycle populations, not only S phase.

Response: Thank you, this control was included in the revision (shown below, on the left, and in supplementary figure 1E of the revision).

Supplementary figure 1E: Top, a cell cycle distribution of control and MCM2-S139A cells with and without exposure to IAA. Bottom, quantification of at least 3 biological replicates.

Comment: Make a statement about expression levels of mutants vs WT and vs endogenous protein. Do so for all other cell lines comparing mutants and WT

Response: Thank you, this information was included in the revision (for example, see revised figure 1 legend).

Comment: - S1B:

Give numbers of all cell cycle populations, not only S phase.

Response: Thank you, this information was included in the revision (on the left and revised supplementary figure 1A).

Supplementary figure 1A. Cell cycle distribution of WT^{SIRT1} and Mut^{SIRT1} cell populations, measured by flow cytometry. Top, a representative cell cycle distribution pattern using an EdU incorporation/DAPI assay. Bottom, quantification of cell cycle fractions. Shown are average fractions derived from 3 biological replicates.

Comment: Make a statement about expression levels of transgenes.

Response: Thank you, this statement was included in the revised legends.

Comment: You call cells ‘isogenic’. I am not sure this expression is appropriate as it involves that transgenes were integrated in the same locus.

Response: Thank you for pointing this out. We agree, and we avoided the use of the word “isogenic” in the revision.

Comment: - S1C:

PPase shows dependency on phosphorylation. Use the A mutant to show that ab detects specifically the S89 site.

Supplementary figure 1F, top panel: Characterization of the custom-made pRecQL4 antibody used in the study. Top, whole cell fractions of two independently isolated clones of HCT116 cells harboring intact RecQL4 (WT 1 and 2) or HCT116 cells harboring RecQL4^{S89A} (S89A 1 and 2) immunoblotted using the pRecQL4-S89, total-RecQL4 and H3 antibodies, demonstrating the specificity of pRecQL4-S89 antibody.

Response: Thank you, this control (using two different stable clones) was included in the revision (above, left, and supplementary figure 1F, top panel).

- Comment: S2A:

Test pre-RC-dependency of signals.

Response: Thank you, as mentioned above, we performed the MCM2-mAID control suggested by the reviewer to test all chromatin immunoprecipitation signals and the result was included in the revision (revised figure 1D, shown the left).

Figure 1D, left: The abundance of pMCM2-S139, MCM2, Treslin, MTBP and pRecQL4-S89 in whole cell and chromatin extracts from HCT116 cells harboring MCM2-mAID along with MCM2-WT and MCM2-S139A with and without auxin (IAA, 16hr).

Comment: - 2E:

Discuss in more detail the association of pRecQ4-S89 with early dormant origins in mid and late time points. This association must be independent of pre-RCs, because there are no pre-RCs anymore on early origins.

Response: We thank the reviewer for raising this important point. In the revision, we discuss the fact that pRecQL4-S89 binding to dormant origins occurs post-replication in the Results section (page 9, last paragraph) and in the Discussion (page 17, top paragraph).

Comment: - 2F:

This figure should involve a potential binding of the phospho-RecQ4 to origins DNA in the absence of pre-RCs as it occurs at least on early dormant oris at late time points. Are there 2 modes of origin binding, pre-RC-dependent and independent?

Response: Thank you, this is an important point. Our observations strongly suggest that pRecQL4 plays a distinct role in the regulation of DNA synthesis, and that in some cases, pRecQL4-origin binding occurs post-replication, hence without a direct interaction with pre-RC. In the revision, this possibility is discussed in the Results section (page 9, last paragraph) and in the Discussion, along with the role of RecQL4 in DNA repair (starting on the last paragraph on page 18).

Comment: - Fig 3:

Mention how the 108A cell line was made (albeit for a previous project). Also describe a bit

better some basic characteristics: how well does it replicate and how efficient which origins fire in normal and replication stress conditions.

Response: Thank you, this issue was also raised by reviewer #1, and we have added a detailed description of our findings from the previous study in the Introduction (page 4, top paragraph).

Comment: The authors please consider to show the origin analysis by ChIP (Fig 4) before the complex formation experiment in Fig 3E. I find this part very hard to follow.

Response: Thank you, we followed this advice, a revised version of former figure 3E is shown as figure 4E of the revision.

Comment: - Fig 3B:

Do dormant origins fire despite the absence of RecQ4-pS89 in Mcm2-S108A cells (no Ex527)? Mention and discuss. If they do not fire this means that absence of phospho-RecQ4 is not sufficient for dormant origin firing.

Response: We thank the reviewer for raising this important point, we agree. Dormant origins are not initiating replication in MCM2-S108A cells, suggesting that the absence of pRecQL4 is not the sole feature distinguishing dormant and baseline origins. In the revision, we discuss this point (pages 10, end of the first paragraph and 18, top paragraph).

Comment: The ATR-Chk1 axis is known to restrict dormant origin firing. However, the authors suggest that ATR is required for Mcm2-S108-mediated dormant origin firing. Is Mcm2-S108 phosphorylated ATR-dependently in this set-up? This issue needs to be mentioned and discussed. ATR inhibitor could be used to test S108 phosphorylation by ATR. ATRi in combination with Mcm2-S108A could be used to test if ATRi-mediated dormant ori firing is also Mcm2-S108-dependent.

Response: We thank the reviewer for raising this point. MCM2-S108 is indeed phosphorylated by ATR, as was shown in our 2022 NAR paper (reference 15), validating the observations of a previous study by the Seidman lab (reference 43). In the revision, we cite both studies and expand the Introduction to clarify this point (top paragraph, page 4).

Comment: 3E:

The experiment is insufficiently described:

- How was the chromatin fraction made?
- How were pMCM2-associated proteins bound to the ab-coupled beads solubilised for subsequent IPs?
- What were the levels of proteins in the supernatant after PCNA-depletion of replisomes? Show the IP-PCNA-flow through as input for subsequent IPs.

Response: We thank the reviewer for this suggestion, which was also raised by reviewer #2. In the revision, we have included a detailed description in the method section pages 3-4.

Comment: The authors intend to show that MTBP and RecQ4-pS89 occupy separate Mcm2-7 fractions. But they use a phospho-Mcm2-S139 antibody for IP, the eluate of which is used for subsequent IPs. Because Mcm2-pS139 and RecQ4-pS89 were shown to be largely exclusive (Fig 1), anti-Mcm-IP (not phospho-MCM-IP) should have been used to eliminate this bias. The authors please describe this more clearly, including what can be concluded from the experiment in light of my point raised.

Response: Thank you for raising this important point. We followed the reviewer's advice, and the results indeed demonstrated that although we did not detect an interaction between MTBP and pRecQL4 in unperturbed cells, MTBP could bind pRecQL4 following exposure to APH. This experiment is included in the revision (see below and in figure 4E).

Figure 4E (previous page): Replication stress facilitates an interaction between MTBP and pRecQL4. Left, experimental strategy. Right, Chromatin extracts from asynchronous HCT116 cells untreated or treated with APH (10 μ M for 1 hr) were used for sequential immunoprecipitation (IP) with PCNA to deplete progressing forks. The PCNA-depleted fraction was immunoprecipitated with pMCM2 antibodies to isolate pre-replication complexes. MTBP or pRecQL4 antibodies were used for further IPs, and the presence or absence of MTBP or pRecQL4 was detected by immunoblotting.

Comment: Fig. 4B/D:

The ChIP data should be displayed the same way it has been shown in previous figures: separated for baseline and dormant origins like in Fig 1D. I can see that the plots shown here constitute a richer data source. However, this comes at the expense of clarity. If the authors see a benefit, the dot plots can be additionally shown, either in the main or supplementary figure. If the authors choose to show dot plots they please also show other ChIP data additionally as dot plots to keep data display consistent.

Here, only baseline origins are shown. Dormant origins should also be shown in the main figure.

The data shown should be more specifically described in the text. The dot plots are quite complicated to read.

Response: Thank you, we have followed the reviewer’s advice and included the heatmaps in the revision, including data corresponding to dormant origins (shown below and in figure 4C and D of the revision). We agree, however, that the XY plots could provide additional information, and they are shown in Supplementary figure 4A.

Figure 4C: A heatmap showing binding of MTBP (left) and pRecQL4-S89 (right) to baseline and dormant origins in HCT116 cells collected immediately after exposure to APH (APH) or at indicated time points post-APH removal. **Figure 4D:** Quantification of the fractions (percent) of baseline (top) and dormant (bottom) origin sequences binding to MTBP, pRecQL4, or both, at the indicated timepoint post-APH wash.

Comment: Fig. 4C:

The RecQL4 cell lines need to be described characterised in more detail. How were the transgenes integrated? Were clones selected? were expression levels of transgenes similar? Did different clones behave similarly? How well did the cells grow and replicate, particularly the ReqQL4 knock out cells. I am surprised they are alive. Are they strongly growth-compromised?

Response: We thank the reviewer for raising this important point. In the revised Materials and Methods section (Supplementary Information), we have provided a more detailed description of the generation and characterization of the RecQL4 null and derivative cell lines. In the revised Results section, we show growth curves and cell cycle analyses of the clones we examined under unperturbed conditions (as shown below and in supplementary figure 5A and B of the revision).

Although we have also initially expected a stronger phenotype, these observations are in line with previously published studies, which reported that RecQL4 knockouts were viable (PMID: 36871012), and depmap analyses suggesting that RecQL4 is non-essential (<https://depmap.org/portal/gene/RECQL4?tab=overview>).

Supplementary figure 5

Supplementary figure 5A: Growth curves for HCT116 clones depleted of RecQL4 or RecQL4 depleted cells harboring RecQL4^{WT} and RecQL4^{S89A}.

Supplementary figure 5B: Left, cell cycle distribution of single clones of HCT116 cells harboring WT, Null or S89A RecQL4 from panel A. Right, quantification of replicates.

Comment: Replication from dormant origins in RecQ4-89A cells as well as Mcm2-S139 phosphorylation status should be shown to complete the dataset.

Response: We thank the reviewer for this suggestion, which tested our hypothesis regarding the role of ReQL4 phosphorylation. NS-seq demonstrates that both RecQL4-Null and RecQL4-S98A supported replication initiation at baseline and dormant origins alike. These data are shown on the left, and in the revision (Figure 5A; pMCM2-S139 ChIP-Seq was reported in our previous publication for WT cells and was not required for the mutated cells given the clear initiation pattern).

Figure 5A: Nascent strand abundance (left, red) and MTBP association measured by ChIP-Seq (right, green) at baseline and dormant replication origins in unperturbed RecQL4 CRISPR knockout HCT116 cells (Null) complemented with either an intact (WT) RecQL4 or with a phospho-deficient mutant of RecQL4 (S89A). Baseline and dormant origins were stratified as described in the legend to Figure 1A and B.

Comment: Fig 5B:

Labelling unclear: release from a 1h treatment with 10uM APH

Response: Thank you, we corrected the labeling (legend for figure 4 in the revision).

Comment: Fig. S5D:

The connection between telomere length and replication stress must be described in a bit more detail.

Response: Thank you, we have included a more detailed description of the experiment (page 14, last paragraph) in the revision.

Comment: - pS89-RecQ4 was implicated in positive firing regulation (Shin et al. JBC 2019). Please discuss.

Response: Thank you, in the revision we discussed these findings (ref. 28) in the Introduction (page 3, 2nd paragraph).

Figure 1D: The abundance of pMCM2-S139, MCM2, Treslin, MTBP and pRecQL4-S89 in whole cell and chromatin extracts from HCT116 cells harboring MCM2-mAID along with MCM2-WT and MCM2-S139A with and without auxin (IAA, 16hr).

Comment: Fig 5E:

pre-RC-dependency of factor binding to origins was not tested.

Response: Thank you for raising this point. As mentioned above, we performed the MCM2-mAID control suggested by the reviewer to test all chromatin immunoprecipitation signals and the result was included in the revision (revised figure 1D, shown on the left).

Comment: Discuss why pRecQL4-deficient cells have strong difficulties to recover from replication stress. After all they do fire dormant origins so these should rescue forks stalled by aphidicolin. Is it because excess dormant origin

firing increases fork collapse? So this might be a special situation when forks are strongly stalled with high aphidicolin. If forks were slow in low aphidicolin the effect of mutant RecQ4 might not be as strong. Is this worth discussing?

Response: We thank the reviewer for raising this point. Our results suggest that in the absence of pRecQL4, cells initiate replication indiscriminately from both baseline and dormant origins. Upon replication stress, the excess replication forks exacerbate the effects of ssDNA and DNA breaks. In cells that can phosphorylate RecQL4, removal of the agent that causes replication stress will trigger RecQL4 dissociation from dormant origins and facilitate MTBP and Treslin binding to unreplicated preRCs, which subsequently

complete replication. In cells that cannot phosphorylate RecQL4, such dormant origins are not available to complete replication, disallowing the recovery from transient inhibition of DNA replication. These events, in addition to the increased likelihood of excess concomitant initiation events and collisions with adjacent replication and transcription machineries, cause the extreme sensitivity of cells devoid of pRecQL4 (either RecQL4 null or RecQL4-S89A) to transient inhibition of DNA replication. We are now testing this hypothesis, and the results are beyond the scope of the current submission. We have modified the model (shown below and in figure 6 of the revision) and expanded the Discussion (page 17, 2nd paragraph, in the revision) to address this point.

Figure 6: A model illustrating dynamic interactions at replication origins during the recovery from replication stress. Left, in cells harboring WT-RecQL4, recruitment of the Treslin-MTBP complex marks a sub-group of baseline origins (e.g. origin 1) whereas pRecQL4-S89 (pRecQL4) associates with dormant origins (e.g. origin 2). When cells encounter replication stress, baseline origins can stall (red halo) but dormant origins, which do not initiate replication, are not affected and still maintain MCM complexes (green ring). Although pRecQL4 binding does not allow initiation from dormant origins under normal circumstances (see figure 2F), RecQL4 dissociates from origins when cells recover from replication stress and allows MTBP re-association with those origins. Upon the binding of MTBP and Treslin to dormant origins, these origins initiate replication to complete DNA synthesis. Right, in cells that do not harbor RecQL4-S89 (either RecQL4 depleted or harboring the RecQL4-S89A substitution), MTBP and Treslin associate with both origins 1 and 2, and replication initiates from both origins when cells encounter acute replication stress. Under these conditions, replication from dormant origins cannot rescue the damage after replication stress is removed, preventing normal recovery and leading to the accumulation of ssDNA and subsequent DNA damage.

In conclusion, we would like again to express our appreciation of the insights provided by all 3 reviewers and thank them for the thorough reading and helpful advice.

Point-by-Point Response to Reviewers' Comments:

Reviewer #1 (Remarks to the Author):

Comment: The authors have made a huge effort to address all reviewers' comments. I think the manuscript offers a significant advance. It is complex, but this reflects the methodology.

I find that the authors have addressed both my and Reviewer number 2's critique in depth. The authors show in several experiments that a single point mutation in the MCM complex blocks the binding of the Treslin-MTBP complex to chromatin. I can't see a flaw in their experimental design. That reviewer number 2 is surprised is irrelevant, it's what the reviewers show, and it appears definitive to me.

Their model is that pRecQL4 distinguishes between dormant and baseline origins and their data supports this model. Specifically, pRecQL4-S89 clearly binds dormant origins in the presence of SIRT1. This is repeated in Figure 2E. Clearly pRecQL4-S89 is binding a subset of origins. In Figure 3B, this binding is shown to be dependent on MCM2 S108, which is a known phosphorylation site. I do not see any ambiguity in these datasets. I could still question what the baseline and dormant origin sequences are, but pRecQL4-S89 binding defines the "dormant" origins precisely. Same for Figure 4C, 5A, 5B.

Response: We thank the reviewer very much for this thoughtful evaluation. We agree.

Comment: Figure 4E: can the authors write 2nd, not 2st.

Response: Thank you for noticing, we corrected the error.

Comment: There is an enormous amount of data in the paper. The authors have done several additional experiments after review. There is no suggestion that the data is contrived or uncontrolled. It's possible their interpretation is wrong or confounded by something we don't understand, but I don't think there is any justification to reject the paper.

Response: Again, we are very grateful for this thoughtful evaluation. We appreciate the reviewer's time and help.

Reviewer #2 (Remarks to the Author):

Comment: The authors have not addressed my concerns. Others have shown that Treslin-MTBP binds in G1 to chromatin and gradually leaves during S-phase.

Response: We thank the reviewer for taking the time to re-review the manuscript. The concerns outlined in the previous critique were addressed in revised Figures 1D-E, 2D, 4E and supplementary figures 1G, 2A-G, 3A, as described in detail in pages 2-11 in the previous point-by-point response to the reviewers. Specifically addressing the reviewer's point above, the previous revisions included new data demonstrating that MTBP, Treslin and pRecQL4 are recruited to chromatin at a low level in a diffuse pattern during the G1 phase and converge to localize primarily at replication origins during S phase, dissociating from origins post-replication, in line with the findings mentioned above by the reviewer.

Comment: It is also very surprising that single point mutations in the MCM complex would block the binding of the Treslin-MTBP complex (about 800 kD) to chromatin.

Response: We respectfully differ with the reviewer's opinion about the potential impact of point mutations and agree with the thoughtful opinion expressed by reviewer #1. Known instances of pervasive effects of point mutations on the activity of large, multi-protein complexes in the context of chromatin include, among others, single-point mutations in components of the Polycomb Group complex, which influence the recruitment of the complex to chromatin with widespread consequences to the transcriptional landscape, and missense mutations in components of the SWI/SNF (BAF) chromatin remodeling complex, which disrupt critical interactions within sub-

complexes with severe consequences in neurodevelopmental diseases. We and others (including the our previous publication, ref #15, and the preprint from the Debatisse group, cited below and in the revised paper, ref #56) observed that DDK regulates replication initiation and that the phosphorylation of MCM2 on a DDK target is essential for replication initiation. These observations are in line with our current report, demonstrating that point mutations at MCM2 phosphorylation targets modulate the binding of the Treslin-MTBP complex to chromatin.

Comment: There is not a good explanation for how pRECQL4 would distinguish between dormant and baseline origins. The overall model seems implausible. The Debatisse group has proposed another model for the control of dormant origins involving regulation of TopBP1 that in principle seems more logical (Koundrioukoff et al., bioRxiv, <https://doi.org/10.1101/2023.11.29.569233>). I would recommend rejection of this manuscript.

Response: As pointed out by Reviewer #1, our observations strongly support a model whereby pRecQL4-S89 binds dormant origins and prevents them from interacting with MTBP-TICRR/TRESLIN. The model outlined in the cited preprint (which cites our previous published work) is consistent with our model, as MCM2-S108, shown in the manuscript to affect recognition of dormant origins, is a substrate of ATR, which is implicated in that model. The two models are not mutually exclusive and we are looking forward to a possible synthesis of these findings in future studies.

Reviewer #3 (Remarks to the Author):

Comment: The revisions are excellent and I fully support publication of the manuscript.

Response: We thank the reviewer very much for the thorough reading of our manuscript and the constructive comments.

Comment: minor points:

line 89: Add 'Although RecQL4 is essential for replication origin firing in Xenopus egg extracts and chicken DT40 cells...'

Response: Thank you, added.

Comment: Fig2F/line 284: '...it is likely that pRecQ4 binding occurred after those dormant origins were passively replicated...' The model in 2F seems inconsistent to me with this sentence as 2F shows RecQ4 bound to pre-RCs. Please clarify.

Response: Thank you, the model shows that pre-RCs that bind pRecQL4 do not bind MTBP-Treslin. To clarify this point, we added a sentence "These observations suggest that replication origins that bind pRecQL4-S89 do not associate with the MTBP-Treslin complex, consistent with origin dormancy."

Comment: Fig S1D: A colour code seems to be missing in the Diagram. I do not see which circle belongs to which condition.

Response: Thank you, we added the color code.

Comment: Fig S3B: Align labelling in right panel with the lanes.

Response: Thank you, we have modified as suggested.

In conclusion, we would like again to express our appreciation of the insights provided by all the reviewers and thank them for the thorough reading and helpful advice.